# Investigating Memory in Model-Free RL with POPGym Arcade

Zekang Wang [* 1]  Zhe He [* 1]  Borong Zhang [1]  Edan Toledo [2]  Steven Morad [1]

## Abstract

How should we analyze memory in deep RL? We introduce tools for analyzing policies under partial observability and revealing how agents use memory to make decisions. To utilize these tools, we present POPGym Arcade, a collection of Atari-inspired, hardware-accelerated environments sharing a single observation and action space. Each environment provides fully and partially observable variants, enabling counterfactual studies on observability. We find that controlled studies are necessary for fair comparisons and identify a pathology where value functions smear credit over irrelevant history. Using this pathology, we demonstrate how out-of-distribution scenarios can contaminate memory, perturbing the policy far into the future. Our code is available at `https://github.com/bol` `t-research/popgym-arcade`.

## 1. Introduction

Decision making often requires reasoning under *partial observability*, where important information is hidden from the agent. This is a challenge for reinforcement learning (RL), which excels in fully observable settings (Silver et al., 2016). To mitigate partial observability, we store and recall sensory information using some form of *memory*. We often evaluate memory models by comparing returns across partially observable tasks. However, deep RL experiments are sensitive to model configurations, observation size, task difficulty, and other confounding factors; especially when combined with memory (Mania et al., 2018; Pleines et al., 2025). This sensitivity makes it difficult to isolate the impact of memory on observability, fairly quantify memory performance, or understand failure modes (Jordan et al., 2024). To address this gap in understanding, we introduce

---
[*]Equal contribution [1]Faculty of Science and Technology, University of Macau [2]Centre for AI, University College London. Correspondence to: Steven Morad <smorad@um.edu.mo>.

*Proceedings of the 43rd International Conference on Machine Learning*, Seoul, South Korea. PMLR 306, 2026. Copyright 2026 by the author(s).

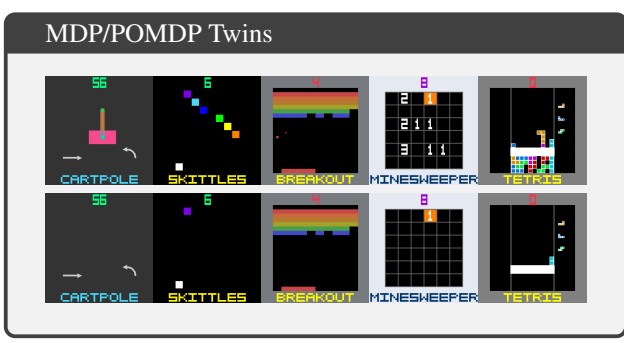

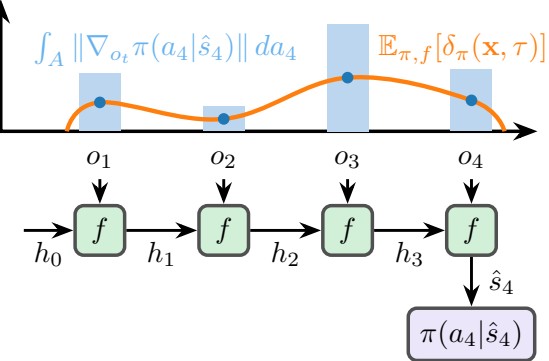

*Figure 1.* A few of our MDP/POMDP twins, enabling counterfactual studies of memory. We also provide memory analysis tools, such as the recall density function (**orange**), which aggregates memory accesses (**blue**) over trajectories to understand how agents use memory ($h$).

tools to isolate and study memory and observability. To utilize these tools, we introduce POPGym Arcade (Fig. 1), an Atari-style hardware-accelerated benchmark of fully and partially observable environment *twins*. Combining twins with our memory analysis tools, we perform counterfactual studies on observability, motivating a secondary study that uncovers memory-based credit attribution failures.

### Contributions

(**C1**) Four tools to measure and interpret memory in RL

(**C2**) Paired MDP/POMDP twins for controlled studies

(**C3**) Evidence that memory introduces bias and confounds return-based comparisons

(**C4**) Discovery of a pathology where memory models incorrectly assign value to irrelevant history

(**C5**) Demonstration of recurrent state contamination, where (**C4**) causes irrelevant past to corrupt a policy

## 2. Preliminaries

**Reinforcement Learning**  A Markov Decision Process (MDP) (Sutton & Barto, 2018) is a tuple $\mathcal{M} = \langle S, A, P, R, \gamma \rangle$, with the set of states $S$, actions $A$, and transition function $P : S \times A \mapsto \Delta(S)$, where $\Delta(S)$ is a distribution over $S$. MDPs also contain a reward function $R : S \times A \mapsto \Delta(\mathbb{R})$ and discount factor $\gamma \in [0, 1)$. At each timestep, we receive a state $s_t \in S$ and select an action $a_t \in A$ from a policy $a_t \sim \pi(\cdot | s_t)$ that takes us to the next state $s_{t+1} \sim P(\cdot | s_t, a_t)$ emitting a reward $r_t \sim R(\cdot | s_t, a_t)$. The Markov property states that $s_t$ contains sufficient information for an optimal policy. In RL, we learn the policy $\pi : S \mapsto \Delta(A)$ that maximizes the expected return $J(\pi, \mathcal{M}) = \mathbb{E}_{\pi, \mathcal{M}}[\sum_{t=0}^{\infty} \gamma^t r_t]$ for $\mathcal{M}$.

**POMDPs**  In Partially Observable Markov Decision Processes (POMDPs), the agent cannot access environment state, instead perceiving the state through noisy or limited sensors. A POMDP $\mathcal{P} = \langle S, A, P, R, \gamma, \Omega, O \rangle$ extends the MDP with set of observations $\Omega$ and observation function $O : S \mapsto \Delta(\Omega)$. At each timestep, the agent indirectly accesses the state through a non-Markov observation $o_t \sim O(s_t)$. Fortunately, the trajectory $\mathbf{x}_t = (o_0, a_0, o_1, a_1 \ldots a_{t-1}, o_t)$, is itself Markov (Sutton & Barto, 2018). A Reward Memory Length (RML) (Ni et al., 2024) of $k$ expresses that the reward depends on only the latest $k$ elements of the trajectory $r_{t+1} \perp \mathbf{x}_t, a_t | \mathbf{x}_{t-k:t}, a_t$. Our resulting new RL objective depends on a memory model $f$ (explained below) and POMDP $\mathcal{P}$: $J(f, \pi, \mathcal{P}) = \mathbb{E}_{\pi, f, \mathcal{P}}[\sum_{t=0}^{\infty} \gamma^t r_t]$.

**Memory and Recurrences**  A memory model $f$ produces a fixed-size latent Markov state $\hat{s}_t \in \hat{S}$ from the trajectory $\mathbf{x}_t$. This confines the complexities associated with partial observability to $f$, allowing the policy to operate on a latent Markov state $\hat{s}$. For conciseness, we introduce the joint observation-action space $X = (\Omega, A); x_t = (o_t, a_{t-1})$. A recurrent memory model $f : X \times H \mapsto \hat{S} \times H$ maintains a summary of the trajectory in the recurrent state $h_t \in H$ and outputs a latent Markov state $\hat{s}_t$ for the policy

$$\hat{s}_t, h_t = f(x_t, h_{t-1}) \qquad a_t \sim \pi(\cdot | \hat{s}_t). \qquad (1)$$

## 3. Related Work

**POMDP Benchmarks**  Progress in RL builds upon benchmarks that provide MDPs and POMDPs (Appendix H). Benchmarks include general-purpose tasks (Bellemare et al., 2013; Morad et al., 2023a) and domain specific tasks like navigation (Chevalier-Boisvert et al., 2018; Beattie et al., 2016; Kempka et al., 2016), psychological experiments (Fortunato et al., 2019), velocity-masked control tasks (Todorov et al., 2012; Ni et al., 2022), and party games (Pleines et al., 2022). Unfortunately, the large

cost of training memory models combined with the sample inefficiency of RL forces researchers into a tradeoff between simplistic low-dimensional observation spaces and a reduced number of experiments to back their claims. Recent hardware-accelerated benchmarks avoid CPU-GPU copies, resulting in significant training speedups (Hessel et al., 2021; Toledo, 2024), and new high-throughput algorithms like PQN (Gallici et al., 2024). We highlight benchmarks from (Matthews et al., 2024a; Pignatelli et al., 2024; Lu et al., 2024; Tao et al., 2025) which offer hardware-accelerated POMDPs.

**Controlled Studies of Partial Observability**  There is ample literature on the theoretical costs and intractability induced by partial observability (Papadimitriou & Tsitsiklis, 1987; Kaelbling et al., 1998; Vlassis et al., 2012; Liu et al., 2022). However, there is less work that empirically quantifies this performance degradation under deep function approximation. Prior work often compares returns between competing memory models (Parisotto et al., 2020; Morad et al., 2023b; Le et al., 2024). With POMDP returns alone, it is not possible to determine whether greater returns are due to more informative latent Markov states or other factors. Paradoxically, literature shows that in certain cases, using memory in MDPs can improve performance, while using memory in POMDPs can reduce performance (Hausknecht & Stone, 2015; Burda et al., 2022; Morad et al., 2023a). For example, a memory model may increase the parameter count of the model, which can improve policy representation capacity (Hansen et al., 2023) without improving the Markov state estimate. On the other hand, optimization of memory-endowed policies is notoriously difficult (Mikhaeil et al., 2022), and a suboptimal memory-free policy may settle in better local optima during training. To isolate the impact of memory, it is necessary to control for all other variables by comparing performance in a partially observable setting against its fully observable counterpart (Jordan et al., 2024). Ni et al. (2022); Morad et al. (2023a); Rajan et al. (2023) provide capabilities for counterfactual studies, but only Ni et al. (2022); Tao et al. (2025) actually perform them. No prior work provides both (1) a collection of tasks with identical MDP and POMDP variants that all share a single observation and action space (twins), and (2) GPU acceleration. Both are necessary to make controlled studies of observability feasible.

**Investigating Memory in RL**  Comparing returns indirectly measures the information content of the latent Markov state $\hat{s}$ created by the memory model. Few works directly answer *how* the model constructs the latent Markov state, providing hints but never complete answers. (Kapturowski et al., 2019) measure the impact of stale recurrent states. (Ni et al., 2024) decouples memory performance from temporal credit assignment, more directly measuring

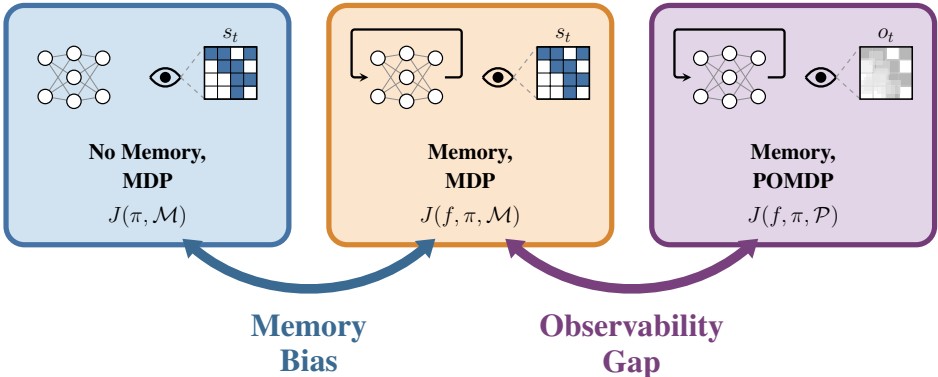

*Figure 2.* **The relationship between the Memory Bias and Observability Gap.** Memory Bias detects hidden confounders introduced by memory model $f$, such as increased parameter count or optimization difficulty. Observability Gap quantifies how well memory mitigates partial observability. Combining said metrics with equivalent MDP/POMDPs ($\mathcal{M}, \mathcal{P}$), enables counterfactual studies of memory.

the quality of $\hat{s}$. Finally, (Elelimy et al., 2024; Morad et al., 2023b) examine the distribution of learned recurrent states.

## 4. Memory Evaluation Tools

The return measures agent capabilities, but does not disentangle memory performance from policy performance, nor does it tell us what the memory model is learning. Below, we describe two tools to isolate and study memory capabilities (Fig. 2), and two more tools to understand what memory models learn.

**Observability Gap** Consider a base MDP $\mathcal{M} = \langle S, A, P, R, \gamma \rangle$. We can induce partial observability by introducing an observation function $o \sim O(s)$, thus creating a corresponding POMDP twin $\mathcal{P} = \langle \mathcal{M}, \Omega, O \rangle$. We are interested in a controlled setting where the observation stream is theoretically sufficient to recover the underlying state, ensuring that the optimal achievable return is identical in both cases.

We focus on the practical difficulty of learning a memory model $f$, to approximate the latent Markov state. To isolate this challenge, we compare the performance of an agent configuration $(\pi, f)$ on $\mathcal{M}$ and $\mathcal{P}$. Under $\mathcal{M}$, $f$ need only pass on incoming states to the policy. The case for $\mathcal{P}$ is more difficult: the agent must infer the Markov state from the trajectory. If $f$ is unable to do so, this will present as a difference in return between $\mathcal{M}$ and $\mathcal{P}$.

**Definition 4.1** (Observability Gap). The Observability Gap measures the performance loss of a memory-based agent $(f, \pi)$ attributable to inferring the state from observations instead of observing it directly

$$\text{Gap}(f, \pi, \mathcal{M}, \mathcal{P}) = J(f, \pi, \mathcal{M}) - J(f, \pi, \mathcal{P}) \quad (2)$$

$$\text{where} \quad \mathcal{M} = \langle S, A, P, R, \gamma \rangle, \quad \mathcal{P} = \langle \mathcal{M}, \Omega, O \rangle. \quad (3)$$

**Memory Bias** As stated in Section 3, prior work hints that introducing memory can produce confounding factors. To measure the inherent performance cost of introducing memory, we propose the Memory Bias. This is the observed performance difference between a memory-endowed policy and its memory-free counterpart on a fully observable task. Factors that cause Memory Bias could include parameter count, optimization complexity, implicit regularization, representational capacity, and more.

**Definition 4.2.** The Memory Bias measures the effect of confounding factors: the performance change caused by the memory model that is attributable to the memory model but **not** attributable to partial observability

$$\text{Bias}(f, \pi, \mathcal{M}) = J(f, \pi, \mathcal{M}) - J(\pi, \mathcal{M}). \quad (4)$$

**Pixel Visualizations** Pixel states and observations enable powerful visualization tools. We implement visualization tools to probe which pixels persist in memory via their impact on the policy. More formally, given a trajectory $\mathbf{x}_n$, we compute latent Markov states $\hat{s}_0, \hat{s}_1, \ldots, \hat{s}_n$ via Eq. (1). Then, we backpropagate through memory and policy, taking the norm of the gradient of $Q$ values with respect to an observation $o_t$ where $t \le n$. We provide variants for both $Q$ learning and policy gradient methods

$$\sum_{a_n \in A} \|\nabla_{o_t} Q(\hat{s}_n, a_n)\|_2^2 = \sum_{a_n \in A} \left\| \frac{\partial Q(\hat{s}_n, a_n)}{\partial \hat{s}_n} \frac{\partial \hat{s}_n}{\partial o_t} \right\|_2^2 \quad (4Q)$$

$$\int_A \|\nabla_{o_t} \pi(a_n | \hat{s}_n)\|_2^2 \, da_n = \int_A \left\| \frac{\partial \pi(a_n | \hat{s}_n)}{\partial \hat{s}_n} \frac{\partial \hat{s}_n}{\partial o_t} \right\|_2^2 \, da_n. \quad (4\pi)$$

This measures how much each pixel from a prior observation $o_t$ propagates through memory and contributes to the $Q$ values Eq. (4Q) or action distribution Eq. (4$\pi$) at time $n$. We experimented with other norms, but we find the $L_2$ norm provides the clearest pixel visualizations.

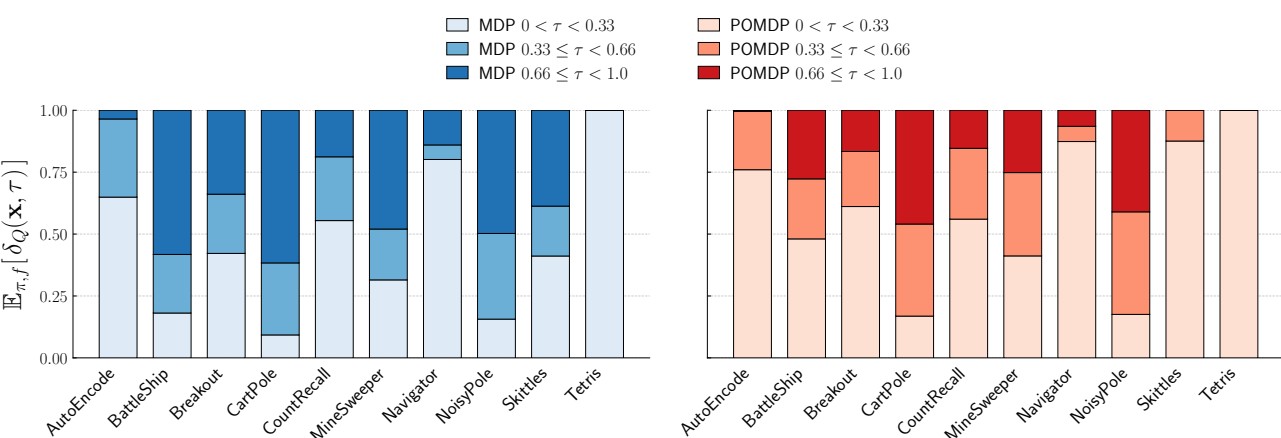

*Figure 3.* **How does the past affect value predictions?** We bin trajectories into thirds $\tau \in [0, 0.33), [0.33, 0.66), [0.66, 1.0)$, and plot the contribution of each bin on the $Q$ value via the recall density $\mathbb{E}_{\pi,f}[\delta_Q(\mathbf{x}, \tau)]$ (Definition 4.3). We aggregate across all memory models and random seeds. All density for MDPs should be in $0.66 \leq \tau < 1.0$, given the Markov property. Instead, we see credit diffusely distributed across trajectories for all models and tasks, demonstrating the value smearing pathology.

**Recall Density** Although pixel saliency (Eqs. (4$Q$) and (4$\pi$)) provides qualitative interpretations of memory, it evaluates single trajectories and may suffer from selection bias, failing to reflect the general behavior of the model. To measure memory usage quantitatively across a distribution of trajectories, we introduce the Recall Density metric. This avoids selection bias and provides a more accurate assessment of how past observations influence the current decision through memory.

To construct this metric, we build upon the gradients defined in the previous section. For a single trajectory $\mathbf{x}_n$, we compute the $L_1$ norm of the observation gradients at each timestep. We then divide each norm by the sum of all gradient norms across the trajectory. This normalization yields the empirical density functions $\delta_Q(\mathbf{x}_n, t)$ and $\delta_\pi(\mathbf{x}_n, t)$. These functions isolate the relative contribution of the observation at time $t$ to the value estimate or action distribution at time $n$.

$$\delta_Q(\mathbf{x}_n, t) = \frac{\sum_{a_n \in A} \|\nabla_{o_t} Q(\hat{s}_n, a_n)\|_1}{\sum_{i=0}^n \sum_{a_n \in A} \|\nabla_{o_i} Q(\hat{s}_n, a_n)\|_1} \quad (5Q)$$

$$\delta_\pi(\mathbf{x}_n, t) = \frac{\int_A \|\nabla_{o_t} \pi(a_n | \hat{s}_n)\|_1 \, da_n}{\sum_{i=0}^n \int_A \|\nabla_{o_i} \pi(a_n | \hat{s}_n)\|_1 \, da_n}. \quad (5\pi)$$

Because agents generate trajectories of varying lengths, we map the absolute timestep $t$ to a continuous, normalized time coordinate $\tau \in [0, 1]$. This allows us to compute the expected memory influence over any set of variable-length trajectories. The Recall Density is the expected value of $\delta_Q$ or $\delta_\pi$, under the trajectory distribution $\mathbf{x}$ induced by the current policy and memory model $\pi, f$. We approximate the Recall Density by taking the sample mean of the empirical densities at the corresponding normalized time points

$$\mathbb{E}_{\pi,f}[\delta_Q(\mathbf{x}, \tau)] \approx \frac{1}{m} \sum_{i=1}^m \delta_Q(\mathbf{x}_{n_i}^{(i)}, \lfloor \tau \cdot n_i \rfloor) \quad (6Q)$$

$$\mathbb{E}_{\pi,f}[\delta_\pi(\mathbf{x}, \tau)] \approx \frac{1}{m} \sum_{i=1}^m \delta_\pi(\mathbf{x}_{n_i}^{(i)}, \lfloor \tau \cdot n_i \rfloor). \quad (6\pi)$$

Here, $\mathbf{x}_{n_i}^{(i)}$ represents the $i$-th trajectory of length $n_i$. We may compute the expectation over all $\tau \in [0, 1]$ to summarize and plot memory recall. One should ensure that $\mathbf{x}$ is not terminal, or risk evaluating $Q$ or $\pi$ at a terminal state.

**Definition 4.3** (Recall Density). The Recall Density measures the expected relative influence of a past observation at the current timestep. For trajectories generated by policy $\pi$ and memory model $f$, it quantifies how much an observation at normalized time $\tau \in [0, 1]$ impacts:

**(4.3.1)** $\mathbb{E}_{\pi,f}[\delta_Q(\mathbf{x}, \tau)]$: The $Q$ value at the trajectory end via $Q$ gradient magnitudes.

**(4.3.2)** $\mathbb{E}_{\pi,f}[\delta_\pi(\mathbf{x}, \tau)]$: The action distribution at the trajectory end via policy gradient magnitudes.

## 5. POPGym Arcade

To utilize our tools, we require a benchmark that both provides MDP/POMDP twins with identical pixel-space states and observation spaces. To this end, we propose POPGym Arcade (Table 1 and Appendix G). Unlike Atari, our tasks utilize (1) stochastic initial states and transition functions (2) hardware acceleration (3) formal POMDP/MDP distinctions (4) known RML and (5) standardized returns to simplify comparisons.

*Table 1.* A summary of our environments. We provide both small (S) and large (L) state/observation spaces for each task. We describe RML over episode length $n$ or a constant length $k$. Frame stacking or fixed-window attention are sufficient to solve $O(k)$ RML POMDPs but not $O(n)$ POMDPs.

| Base Envs | Total Envs | State/Observation Size | Action Space |
|---|---|---|---|
| 10 | 120 | $128 \times 128 \times 3$ $256 \times 256 \times 3$ | $\uparrow, \downarrow, \leftarrow, \rightarrow, \times$ |

| Base Env | Notes & POMDP Utility | RML |
|---|---|---|
| Autoencode | Push and pop to latent stack, NC[1] circuit complexity | $O(n)$ |
| CartPole | Classic deterministic control task, useful for debugging | $O(n)$ |
| CountRecall | Increment and query latent counters, $TC^0$ circuit complexity | $O(n)$ |
| BattleShip | Large configuration space and highly-stochastic transition function | $O(n)$ |
| Breakout | Disappearing ball, constant RML, and long episodes | $O(k)$ |
| MineSweeper | Difficult games rules, requires in-context exploration | $O(n)$ |
| Navigator | Navigation and path planning, requires latent map representation | $O(n)$ |
| NoisyPole | State estimation under perturbations, surrogate for real-world control | $O(n)$ |
| Skittles | Obstacle avoidance with flickering obstacles, requires path replanning | $O(k)$ |
| Tetris | Hard spatial task; human competitions for MDP & POMDP | $O(n)$ |

**Environment Twins** Our twins introduce two notions of state: a low dimensional hidden Markov state $\tilde{s} \in \tilde{S}$ and a pixel Markov state $s \in S$. For example, in MDP MineSweeper, $\tilde{s}$ contains the location of all mines, while $s$ contains pixels representing numbered tiles inferring mine locations. Both states are Markov, but deterministic transitions in $\tilde{S}$ become stochastic in $S$. While $\tilde{S}$ varies across tasks, $S$ is identical for all tasks. By separating $\tilde{s}$ from $s$, we can introduce POMDP variants with an observation function $O : \tilde{S} \mapsto \Delta(\Omega)$. Consequently, twins share identical underlying transition dynamics $\tilde{S} \times A \mapsto \tilde{S}$. All tasks share a singular pixel state/observation space $S = \Omega$ and action space $A$. As a result, we can reuse a single model across all tasks and task configurations. We can even change from POMDP to MDP mid-episode, or train a policy on both MDPs and POMDPs concurrently.

**Task Diversity and Difficulty** All our environments are human-playable and inspired by existing card games, board

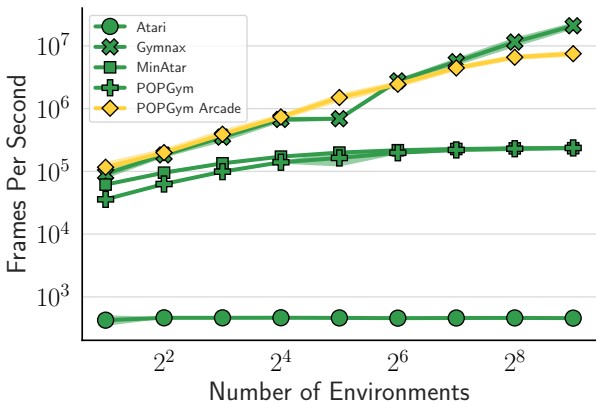

*Figure 4.* **Environment throughput.** We compare the throughput of our environments to other well-known environments, using a GPU and CPU. We parallelize CPU environments using synchronous VectorEnvs, and shade the 95% bootstrapped confidence interval.

games, and video games. We design each task to test a different aspect of memory, described further in Appendix F. We provide a range of difficulties, from trivial to extremely difficult, with mean human scores ranging from 0.00/1.00 to 0.97/1.00 on POMDPs, and an overall POMDP benchmark human average of 0.33/1.00 (Appendix M). Our most difficult task, POMDP Tetris, is played by Tetris Grandmasters at tournaments and has yet to be solved by RL.

**Environment Throughput** We implement POPGym Arcade purely in JAX, enabling parallelism for sample efficient RL. As we demonstrate in Fig. 4, we achieve linear scaling up to $2^7$ parallel environments before saturating the compute units. Even with high dimensional pixel observations, POPGym Arcade achieves throughput approximately 10,000 times faster than the CPU-based Atari environments, and achieves similar throughput to hardware-accelerated Gymnax (Lange, 2022) while generating observations four orders of magnitude larger.

## 6. Results

We investigate two questions. First, we ask if the return is sufficient to evaluate memory, or if confounding factors spoil comparisons between memory models. Second, we perform a examination of how policies utilize memory.

### 6.1. Experimental Setup

In our experiments, we train policies across all environment configurations using five random seeds and seven different memory architectures: a transformer (Vaswani et al., 2017), a recurrent linear transformer (Katharopoulos et al., 2020), a linear test-time training model (Sun et al., 2025),

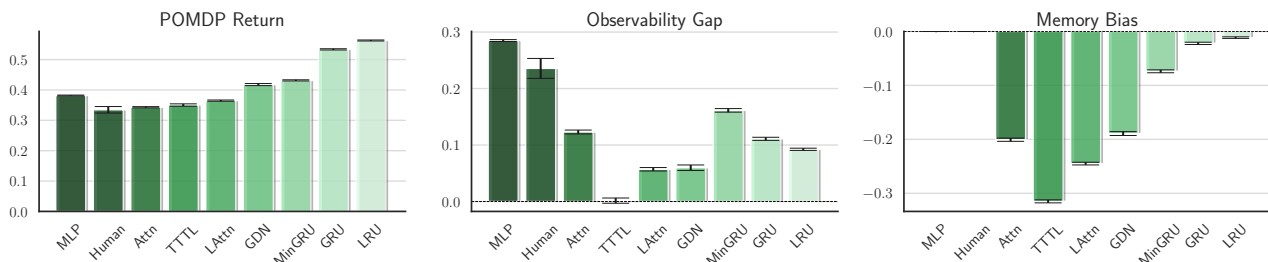

*Figure 5.* **Disentangling the return with our memory analysis tools.** We plot the POMDP returns $\in [0, 1]$, the Observability Gap, and Memory Bias. We aggregate scores over all environments and difficulty configurations. Whiskers represent the 95% confidence interval over five seeds. Differences in Memory Bias between models suggests the return is a confounded metric for benchmarking memory.

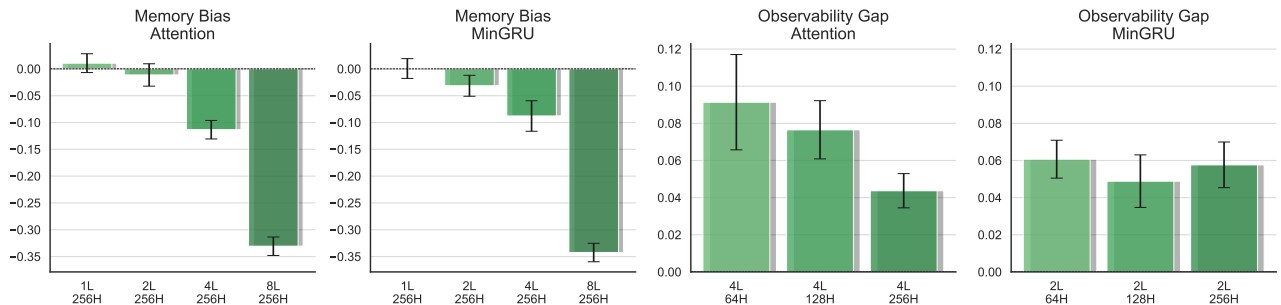

*Figure 6.* **Memory model complexity affects Memory Bias and Observability Gap.** We report the Gap and Bias metrics for various memory models on the BattleShip and MineSweeper environments, computing the mean and 95% CI over three random seeds. We vary the number of memory layers $L$ and recurrent state dimensionality $H$. The Gap and Bias metrics roughly form a Pareto front: increasing memory model complexity (parameter count) tends to worsen Memory Bias but improve Observability Gap. Such metrics can help determine an optimal tradeoff between memory model representational capacity and optimization difficulty.

Gated Delta Nets (Yang et al., 2025), a linearized GRU (Feng et al., 2024), a standard GRU (Chung et al., 2014), and the LRU state space model (Orvieto et al., 2023), all explained in further detail in Appendix I. We note that the standard transformer does not maintain a recurrent state, and therefore cannot persist episode state across training epochs like recurrent models, struggling in long RML tasks (Table 1). To better distinguish between shortcomings in memory models and downstream policies, our models utilize a skip connection that bypasses the memory model, allowing the policy agency to ignore memory when unnecessary, such as in MDPs.

We highlight our results using simple on-policy $\text{TD}(\lambda)$ Q learning (PQN, Gallici et al. (2024)), eliminating the possibility of common confounding factors such as shared value/policy trunks, target networks, and replay buffers. To demonstrate that our findings generalize across algorithms, we repeat most experiments using an on-policy policy-gradient method (PPO, Schulman et al. (2017)) and an off-policy value-based method (DQN, Mnih et al. (2015)) in Appendix C.

### 6.2. Disentangling Returns

What portion of the return can we attribute to memory models closing the observability gap, and what portion can we attribute to other factors? In Fig. 5, we plot the return, Observability Gap, and Memory Bias (Definitions 4.1 and 4.2) aggregated over all tasks and difficulty levels.

We find that Memory Bias varies between models and can spoil return-only comparisons. Recall that both the Observability Gap and Memory Bias operate in the same space as the return (i.e., one can directly compare the scales of the return, Observability Gap, and Memory Bias). For most models, the Observability Gap and Memory Bias have similar scales, demonstrating the weakness of return-only comparisons and the utility of disentangled metrics. Take the MinGRU and GRU, for example. The relative effect of Bias $0.07 - 0.02 = 0.05$ impacts the return as strongly as the Observability Gap $0.16 - 0.11 = 0.05$. Purely negative biases across models points to a memory-induced performance penalty, rather than benefit. Although memory models increase parameter count and can approximate a wider class of functions than feedforward networks, such effects are not strong enough to overcome other negative factors.

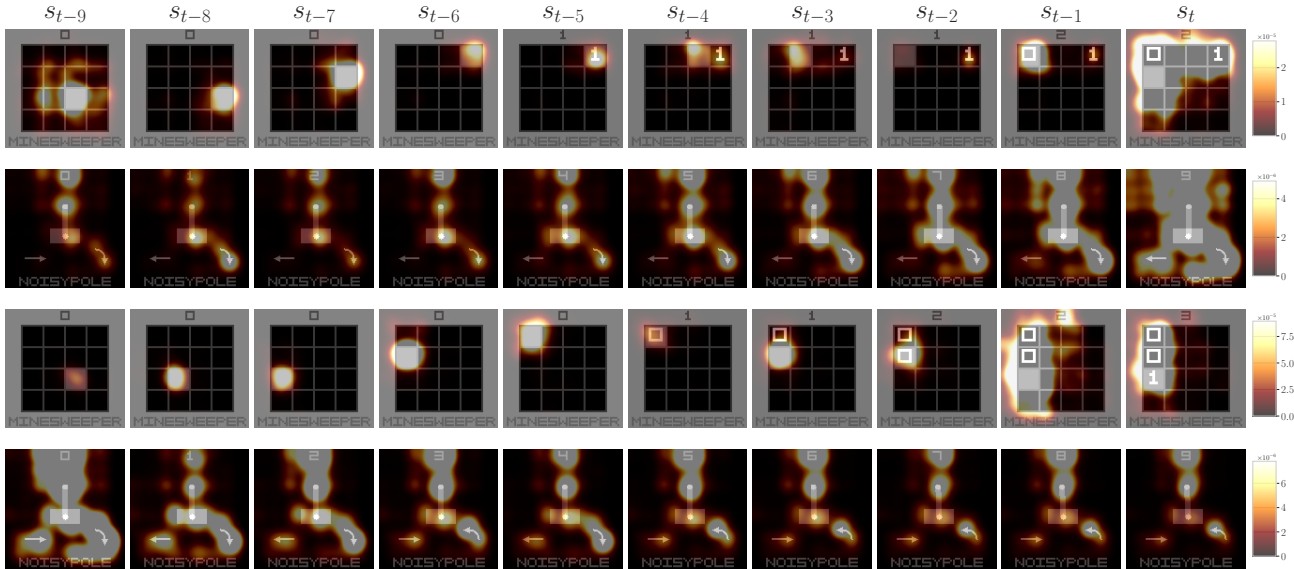

*Figure 7.* **How do trained agents use memory?** We plot pixelwise memory gradients Eq. (4$Q$) for the LRU (top rows) and GRU (bottom rows), denoting how much each pixel contributes to value estimate $V(s_t)$ through memory. In these MDPs, $V_*(s_t)$ is independent of $s_{t-k} \ldots s_{t-1}$, yet memory incorrectly smears value credit over uninformative past states, even with a residual connection bypassing the memory model. Smeared value attribution suggests value estimators may not generalize to new trajectories.

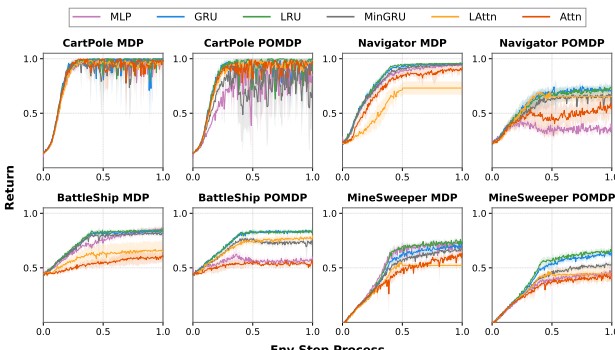

*Figure 8.* **Training stability and convergence.** We plot the mean return and 95% CI over 5 seeds for tasks used in our pathology analyses. Particular models that we focus on (LRU, GRU) appear converged with low variance across seeds, ruling out optimization instability as a confounder for the value smearing experiments.

### 6.3. Metric-Based Interventions

Do the Observability Gap and Memory Bias provide insight into memory model shortcomings and enable interventions? We perform experimental sweeps over recurrent state size and number of recurrent cells in Fig. 6.

Increasing the number of memory cells (depth) beyond a point appears to hurt the Memory Bias metric. A large negative Memory Bias appears when a memory model performs worse on the MDP than a memory-free MLP does. This suggests that the optimizer struggles to find good parameters for the memory model. This could be caused by

impaired gradient flow through memory or a loss landscape with many poor optima. When experiencing a large negative Memory Bias, reducing memory model size seems to be a useful intervention.

Alternatively, increasing the recurrent state size seems to improve the Observability Gap metric. A large Observability Gap results from a model performing significantly better on an MDP than the POMDP twin. We hypothesize that this is due to state aliasing, where the memory model erroneously maps unique trajectories to shared Markov states, perhaps due to lack of representational capacity in the memory model. Increasing the size of the memory model seems to be an effective intervention for large Gaps. Together, the Memory Bias and Observability Gap provide signal for tuning model complexity.

### 6.4. The Value Smearing Pathology

Next, we investigate how our trained policies use memory with our gradient visualization tools from Section 4. After training[1], we perform qualitative pixel-level gradient visualizations on both MDPs and POMDPs (Fig. 7 and Appendix A). We highlight the MDP results because MDPs have a known ground-truth credit distribution: the return depends only on the current state/observation.

---

[1] A prerequisite for analyzing memory pathologies is a set of converged agents. Fig. 8 illustrates that the pathologies we discuss are not artifacts of training instability, but rather a byproduct of the solution learned by converged memory models.

The resulting gradient distributions appear *smeared* over all prior states, even those containing no useful information (e.g., empty board denoting only past actions). We call this phenomena *value smearing*. Value smearing demonstrates that the policies struggle to decorrelate past actions from the forward-looking return, and may signal that the memory model and $Q$ function may have overfit to the trajectory distribution induced by the current policy. This suggests that memory optimization difficulty might be one negative confounder.

One may wonder if these results are cherry-picked. Fortunately, we can use the Recall Density (Definition 4.3) to measure memory access in expectation. In Fig. 3, we examine recall across all tasks, aggregating results across memory model and seed. We find evidence of value smearing across all models and tasks. States from the first two-thirds of an episode ($\tau < 0.66$) impart a significant impact on Q values across all MDPs. See Appendix B for a more granular breakdown by model.

### 6.5. Recurrent State Contamination

Naturally, we wonder what practical consequences arise from value smearing. Perhaps it induces robustness to Out of Distribution (OOD) scenarios. By distributing credit diffusely, a single OOD observation may impart little impact on the policy. To test this hypothesis, we first collect trajectories following learned policies. Then, we perturb these collected trajectories similar to Stone et al. (2021) and analyze how the relative (mean-centered) Q values $A(s, a) = Q(s, a) - |A|^{-1} \sum_{a' \in A} Q(s, a')$ and policy change. We measure the impact of (1) adding noise to a single frame in each trajectory (Fig. 9 and Appendix D), and (2) shuffling the first few observations in each trajectory (Fig. 10). Observation noise is a common problem in POMDPs, but it is unclear whether the memory module or CNN is responsible for any robustness or sensitivity. By shuffling the beginning of the trajectory, any observed effects will be caused purely by memory.

We find that memory-endowed agents are not robust and they do not smooth over OOD observations. Even a single OOD observation can be disastrous for the policy. This is especially worrisome for recurrent models, where the OOD observation can persist in the recurrent state, *contaminating* it and perturbing the policy far into the future. We see similar effects even when we permute the trajectory, removing possible CNN confounders. Our results demonstrate large sensitivity to both OOD observations and trajectories. This is especially concerning for applications where OOD scenarios are expected, such as when applying RL to the real world or using offline RL.

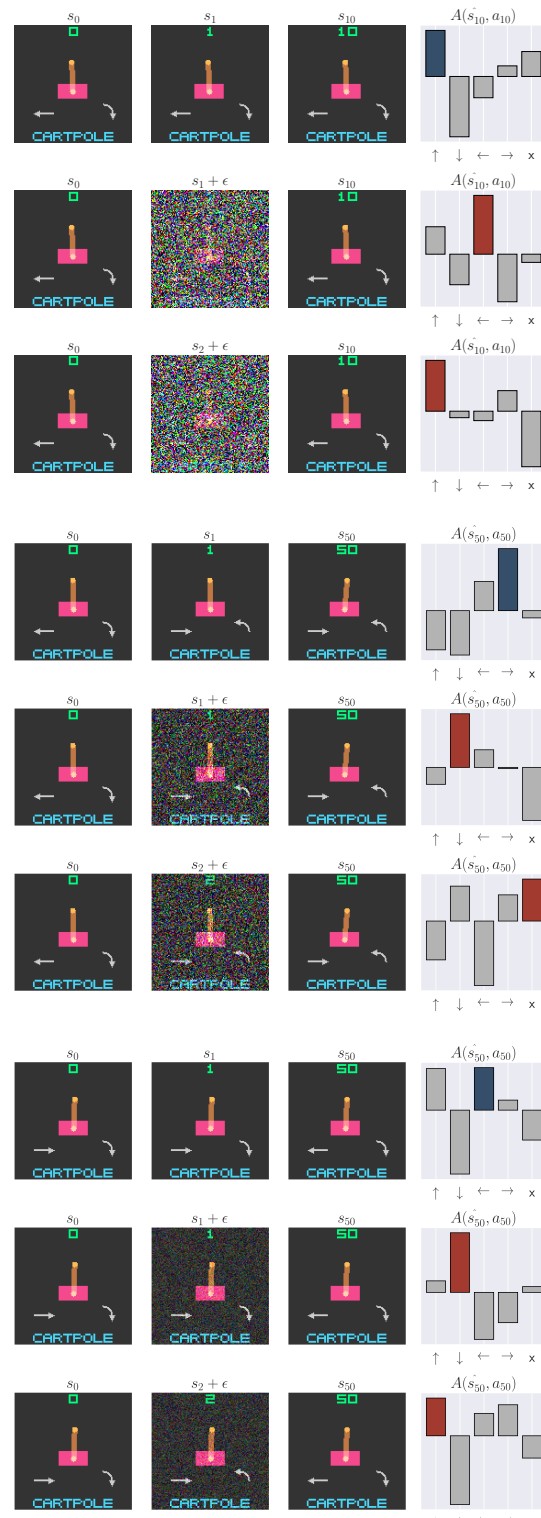

*Figure 9.* **Recurrent state contamination.** We inject varying amounts of noise into one past observation, demonstrating perturbations in relative $Q$ values ($A$) and the greedy action for the LRU-based policy.

# 7. Limitations and Future Work

**Memory Model Comparisons**  Although we provide returns for various memory models, we stress that our goal is not to determine the best memory model. We note that the LRU achieves the best overall return, however this is unlikely to hold across different implementations or configurations. We selected memory model configurations based on our available computational resources, using the recurrent state size as the comparison axis. We could have just as easily chosen parameter count or wall-clock training time as the comparison axis, which could change how the models compare to each other.

**Understanding Memory Bias**  We suggest value smearing as one possible cause for memory-induced performance penalties, but proving causation is difficult and requires further investigation. Future studies should avoid return-only comparisons and utilize the Gap and Bias metrics to measure model capabilities.

**MDPs and State Contamination**  We highlight some of our findings on MDPs because MDPs provide known ground-truth credit assignment over prior states, while POMDPs do not. Equivalent experiments on POMDPs produce similar results, but it is difficult to isolate and quantify credit assignment error in POMDPs.

**Model-Based RL and LLMs**  Our experiments focus on pixel-based model-free RL. We train pixel-based policies from scratch using objectives based on policy-gradient and value-iteration. It is unclear whether value smearing and contamination phenomena appear in model-based RL, such as in world models. Perhaps next-state prediction losses provide stronger learning signal and avoid smearing and contamination phenomena. Furthermore, modern LLMs often undergo heavy RL finetuning. Whether these LLMs exhibit similar smearing or contamination phenomena was not tested, but could be useful for improving LLM performance on long context in-context tasks.

# 8. Conclusion

We introduced a methodology to dissect memory in RL, revealing that return-based comparisons can be misleading. We proposed alternative metrics that account for and measure confounding factors when comparing memory models. Then, we identified a value smearing pathology, where value is incorrectly attributed to irrelevant history. We demonstrated that this credit assignment failure makes memory-endowed policies sensitive to OOD scenarios, corrupting decisions far into the future.

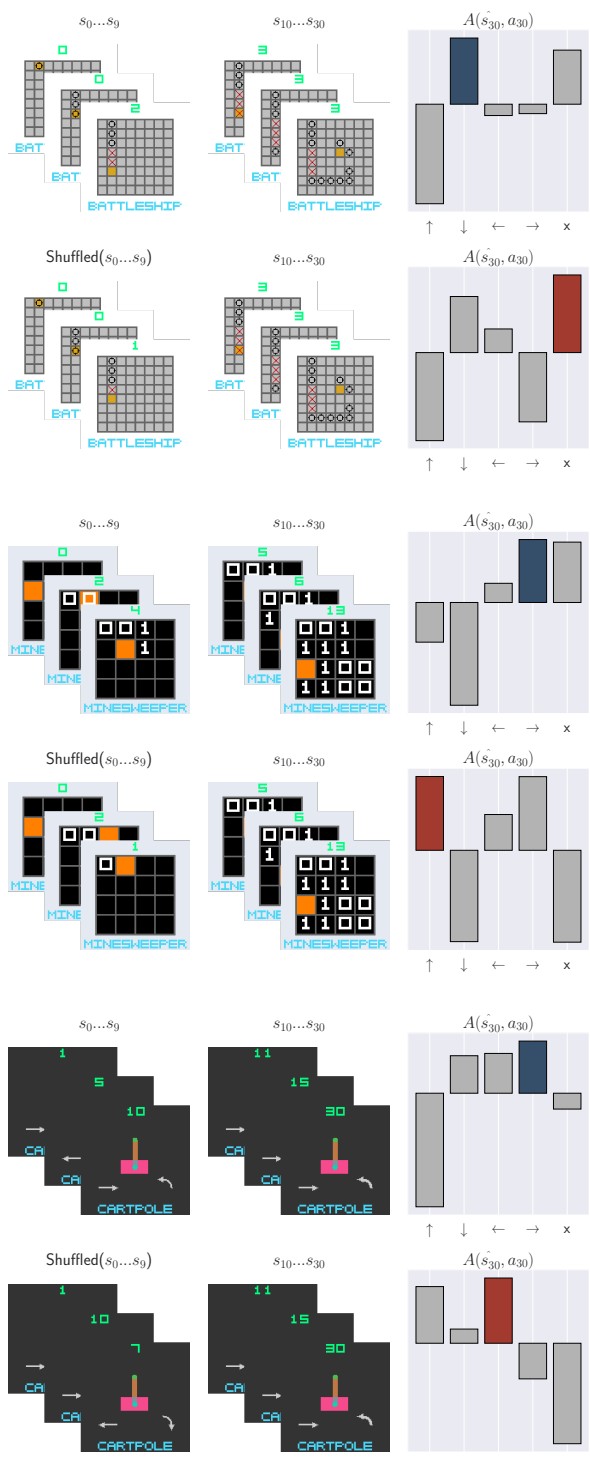

*Figure 10.* **Confounder-free recurrent state contamination.** We shuffle the beginning of each trajectory to remove any CNN confounders. Even in-distribution observations can contaminate the recurrent state when the trajectory is OOD. The BattleShip and MineSweeper examples use the LRU and the CartPole example uses the transformer, demonstrating this effect even for non-recurrent memory models.

## Impact Statement

This paper presents work whose goal is to advance the field of machine learning. There are many potential societal consequences of our work, none of which we feel must be specifically highlighted here.

## Acknowledgements

This research was supported by projects SRG2025-00006-FST and CG2026-FST from the University of Macau. This work was performed in part at the High Performance Computing Cluster (HPCC) which is supported by Information and Communication Technology Office (ICTO) of the University of Macau, and the Super Intelligent Computing Center (SICC) which is supported by State Key Laboratory of Internet of Things for Smart City (SKL-IOTSC), University of Macau, Macao SAR.

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

# Appendix

## A. Additional Pixel-Level Memory Analysis Experiments

We provide further pixel-level analysis of our environment is using Eq. (4$Q$). We see that memory models ignore Markov observations, still storing and recalling information in the MDP case.

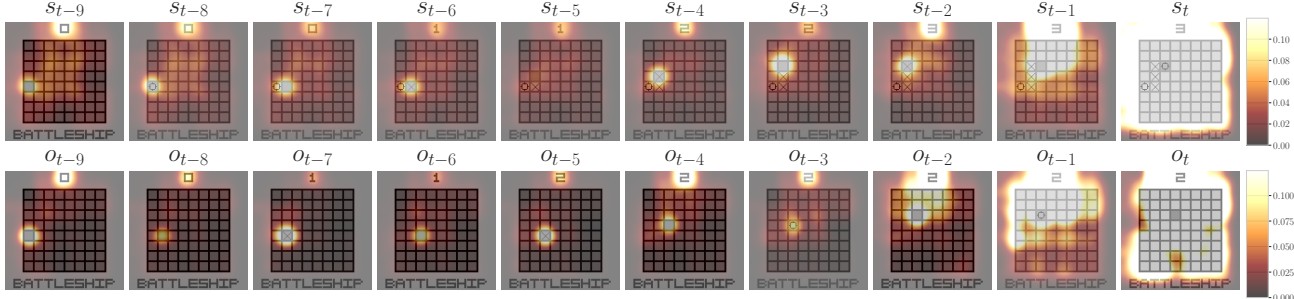

*Figure 11.* LRU saliency on the BattleShip task, plotted via Eq. (4$Q$) using the L2 norm. The top row represents the MDP and the bottom row represents the equivalent POMDP.

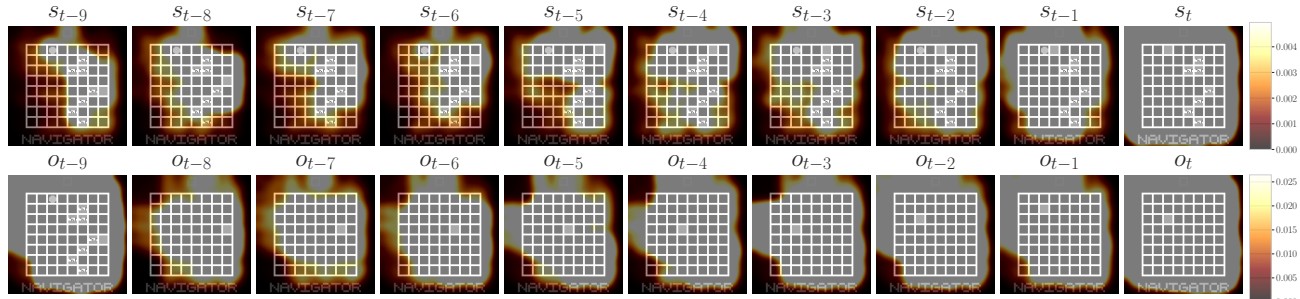

*Figure 12.* LRU saliency on the Navigator task, plotted via Eq. (4$Q$) using the L1 norm. The top row represents the MDP and the bottom row represents the equivalent POMDP.

## B. Recall Density Analysis by Model and Task

In this section, we also plot the recall density for each model and task. We discretize $\tau$ into sixteen bins and compute the density over five random seeds. No matter the memory model, uninformative prior observations still tend to affect future decisions in MDPs. Even in the best cases, much of the density in MDPs is distributed across old observations (Fig. 15).

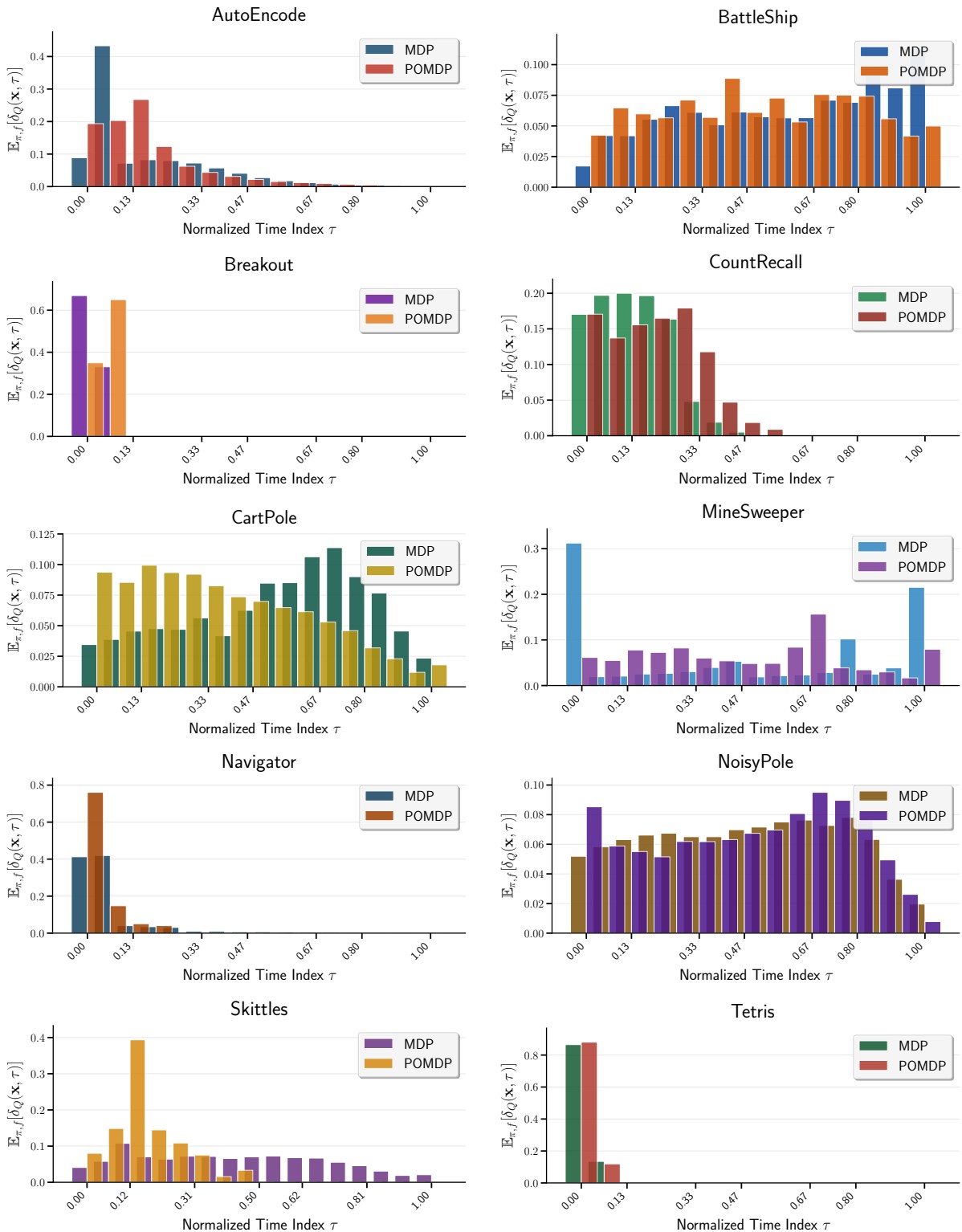

*Figure 13.* Recall density of Linear Transformer categorized by environment, over five random seeds.

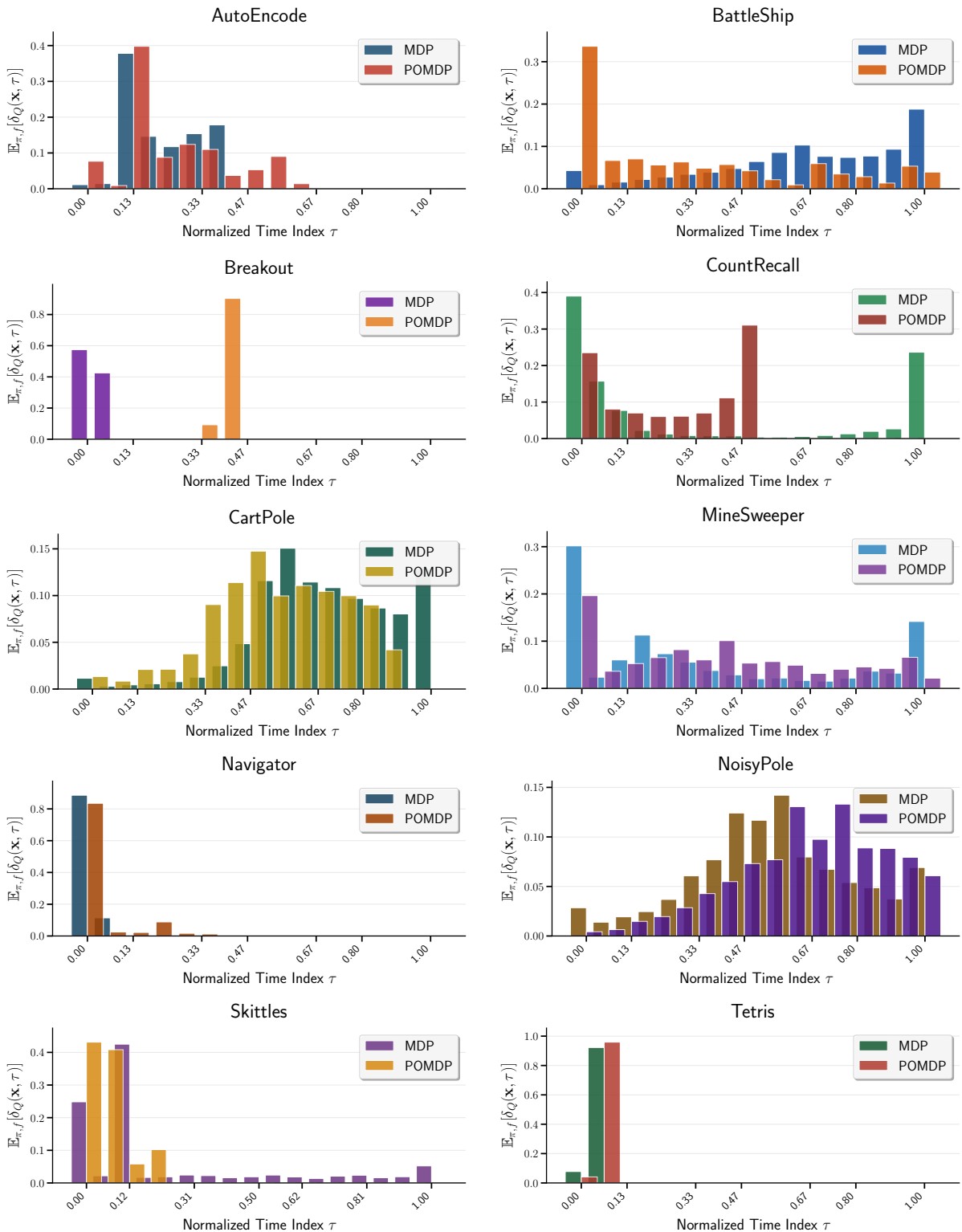

*Figure 14.* Recall density of GRU categorized by environment, over five random seeds.

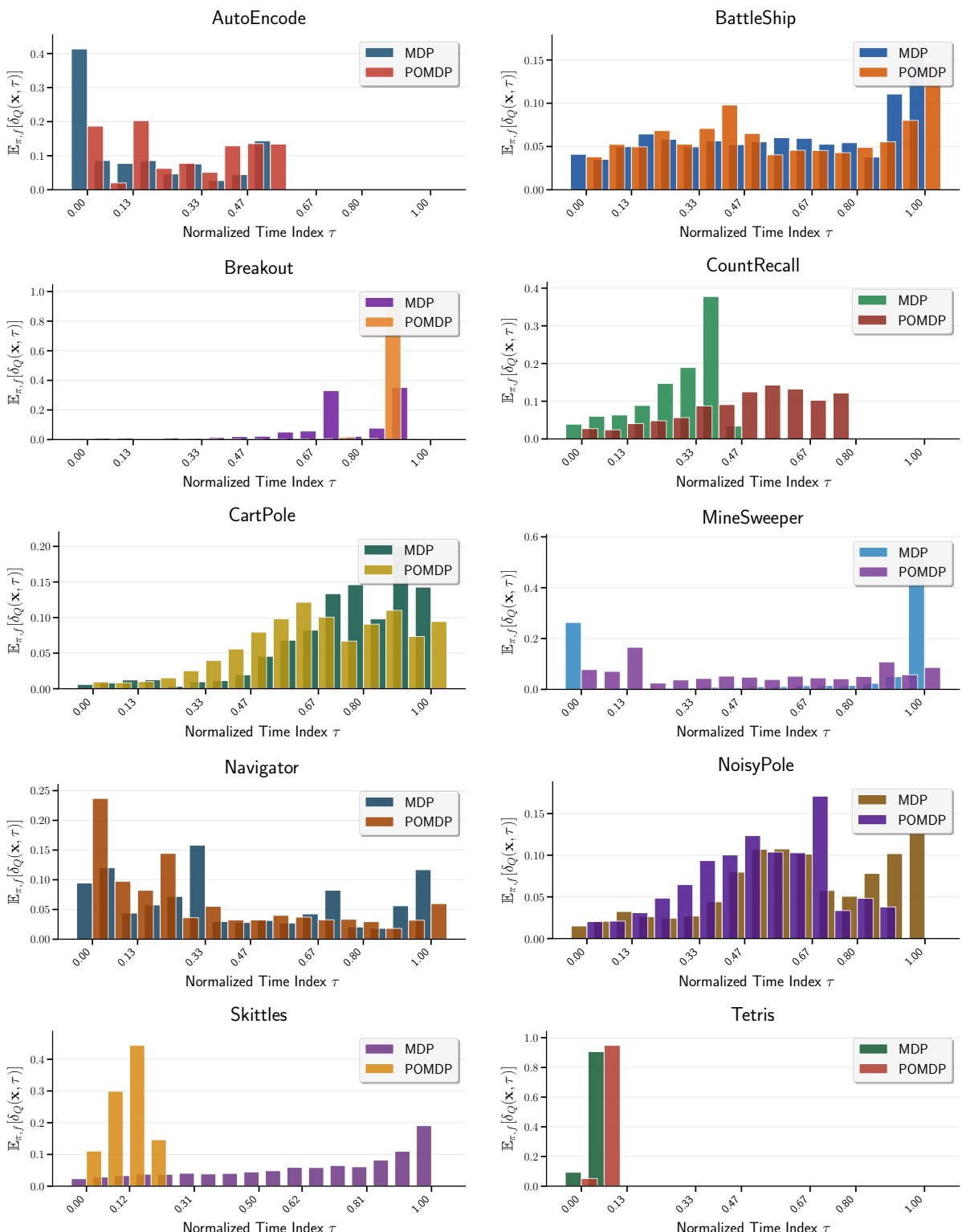

*Figure 15.* Recall density of LRU categorized by environment, over five random seeds.

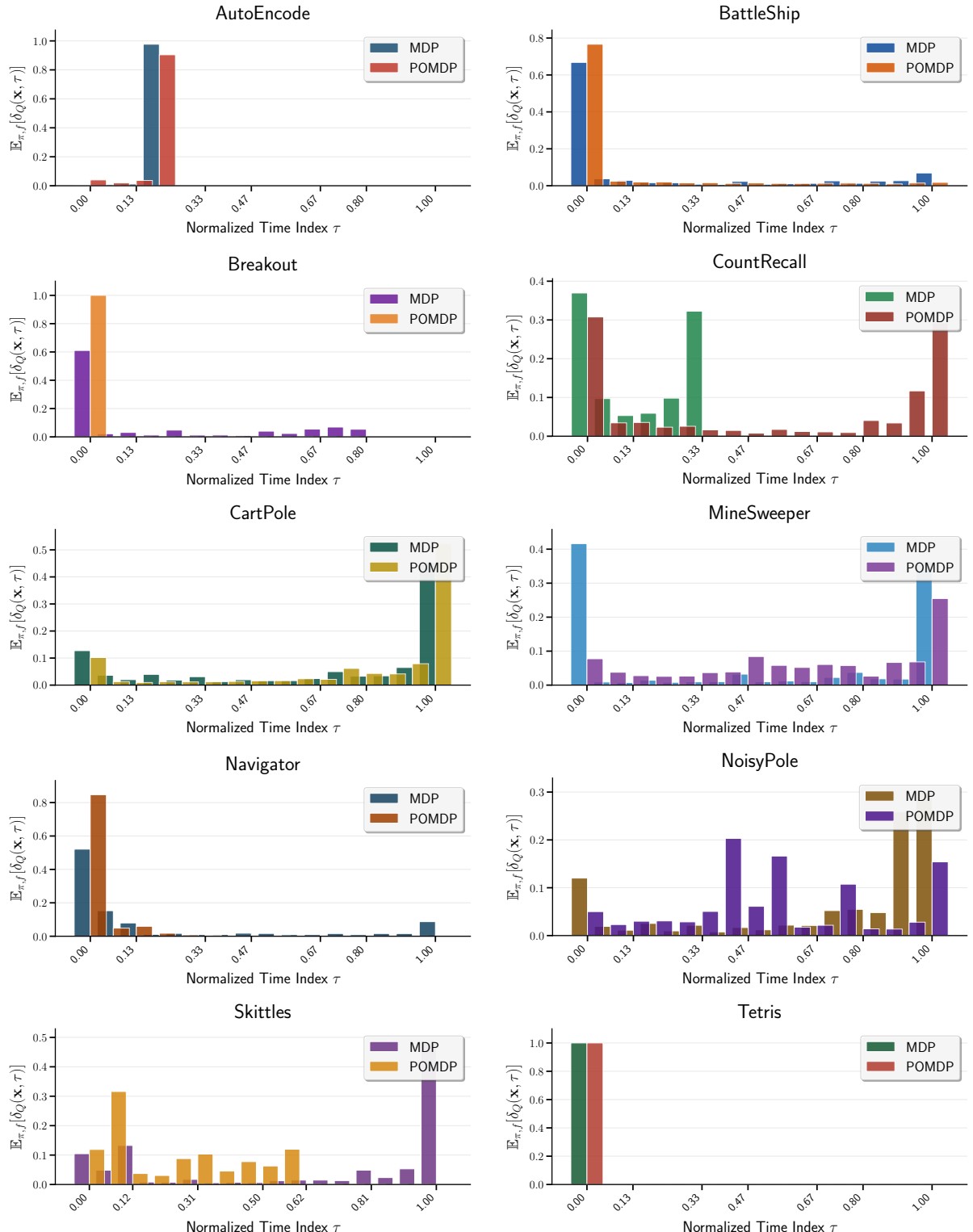

*Figure 16.* Recall density of MinGRU categorized by environment, over five random seeds.

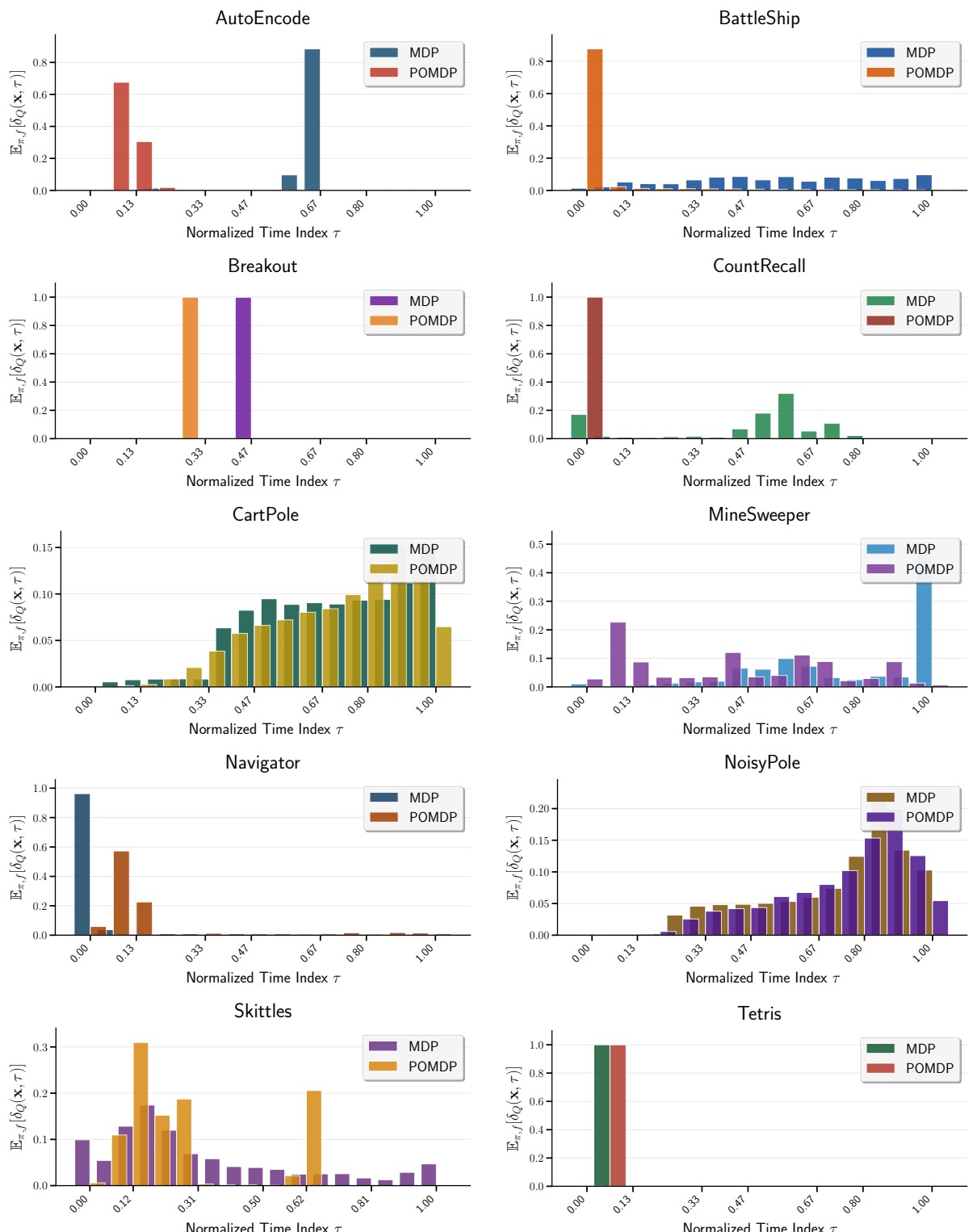

*Figure 17.* Recall density of the transformer categorized by environment, over five random seeds.

## C. PPO and DQN Experiments

PQN is as close as we can possibly get to pure deep value function approximation, discarding replay buffers, target networks, and so on. However, it is not clear if our PQN findings extend to policy gradient algorithms (PPO) or algorithms with experience replay (DQN). In this section, we repeat our PQN analyses for the DQN and PPO algorithms, demonstrating our results generally hold across other RL algorithms as well. Specifically, we aim to demonstrate the generalizability and applicability of Conclusion (**C3**), (**C4**) and (**C5**).

### C.1. Model-Based Reward Predictors

To simulate a mechanism analogous to model-based prediction, we set the discount factor $\gamma$ to 0 in the PQN experiment. This modification restricts the Q-function to learning only the immediate reward, thereby decoupling it from long-term cumulative returns. From the figure Fig. 18, we can clearly observe that value smearing is still present.

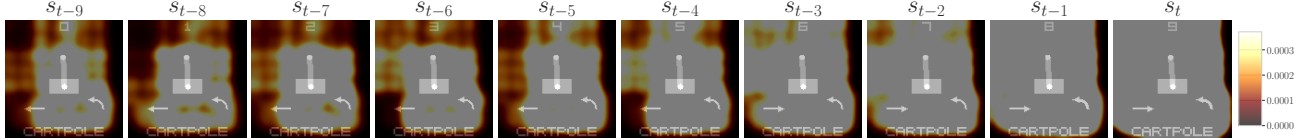

*Figure 18.* LRU saliency on the CartPole MDP task, with PQN and $\gamma = 0$. Setting the discount factor $\gamma$ to 0 is intended to simulate a model-based mechanism, allowing for analysis of how the agent uses historical information to evaluate immediate states or rewards.

### C.2. PPO

To further verify the generalizability of our findings, we extend our experiments to PPO and analyzed the policy's memory utilization. In Fig. 19, We plot our new metrics (Definition 4.1, Definition 4.2) for PPO. Figure 21 and Fig. 20 indicate that our findings extend to policy optimization, where the issue persists but is replaced by policy smearing. Similarly, Fig. 27 provides evidence that the phenomenon described in Item (**C5**) also occurs in PPO.

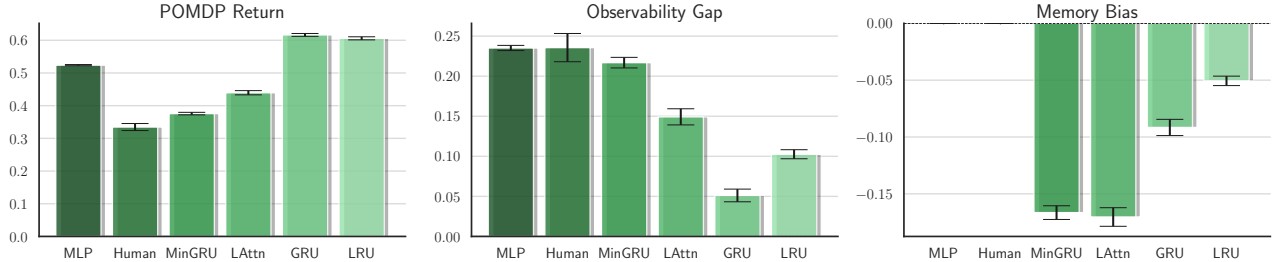

*Figure 19.* Disentangling returns for the PPO algorithm. We replicate the analysis from Fig. 5 with PPO to verify the generalizability of our findings. We plot the aggregated POMDP returns, Observability Gap (Definition 4.1), and Memory Bias (Definition 4.2) across all environments and the Easy difficulty configuration. The results confirm that the negative memory bias and performance gaps observed in value-based methods (PQN) persist in policy optimization settings.

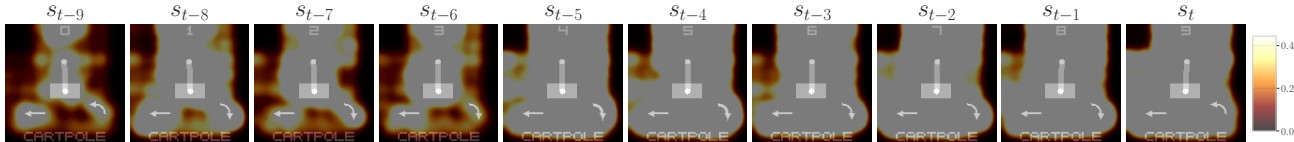

*Figure 20.* Pixel-level saliency for PPO with LRU on CartPole MDP. We visualize gradients of the policy logits with respect to past observations. The results demonstrate that policy smearing occurs in PPO method.

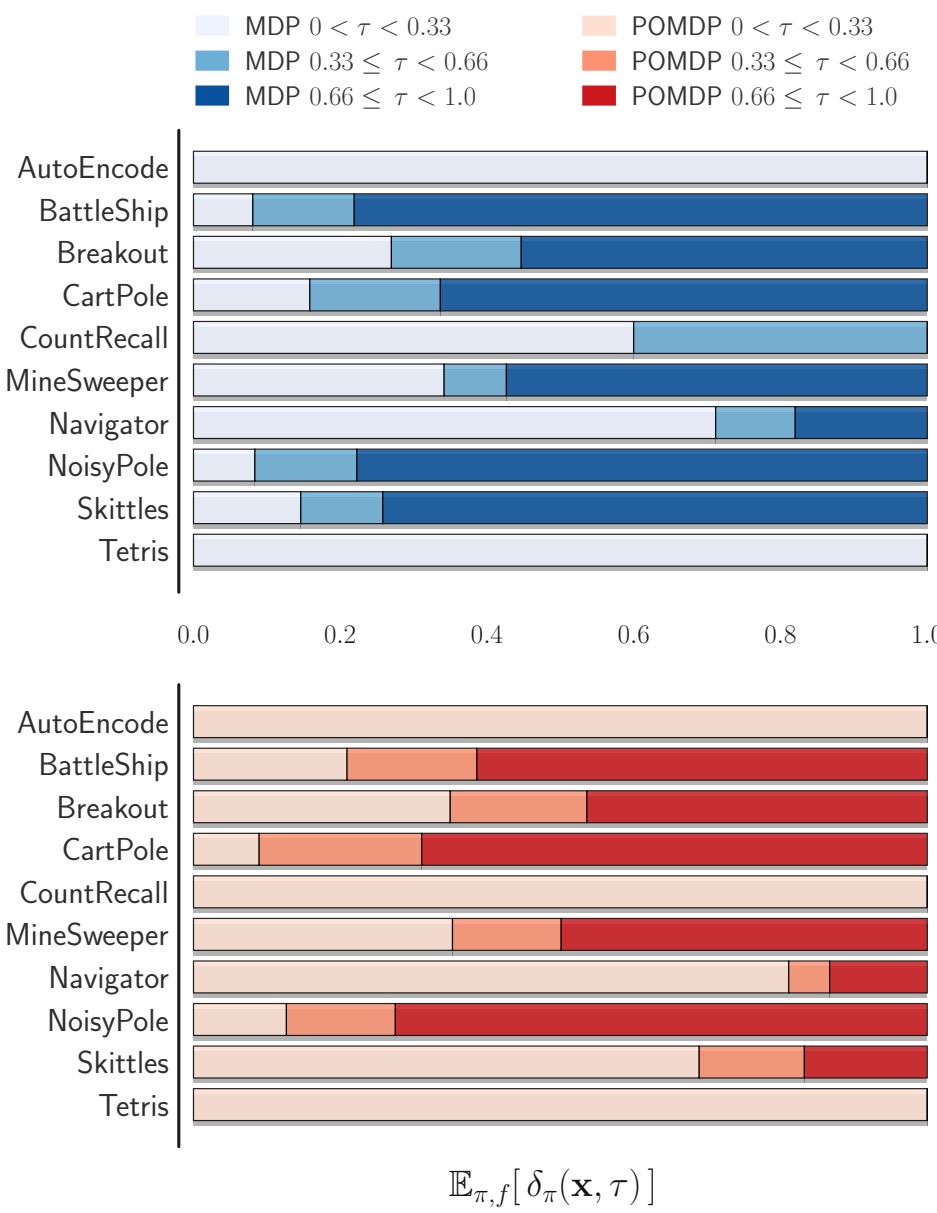

*Figure 21.* Aggregate Recall Density for PPO on Easy task variants. We plot the expected contribution of historical observations to the value estimate. The results confirm that policy smearing exists in PPO.

## C.3. DQN

We extended our verification to RL algorithms equipped with replay buffers. Specifically, we evaluated DQN in three representative environments, with the supporting evidence presented in Fig. 22, Fig. 24, Fig. 25 and Fig. 27.

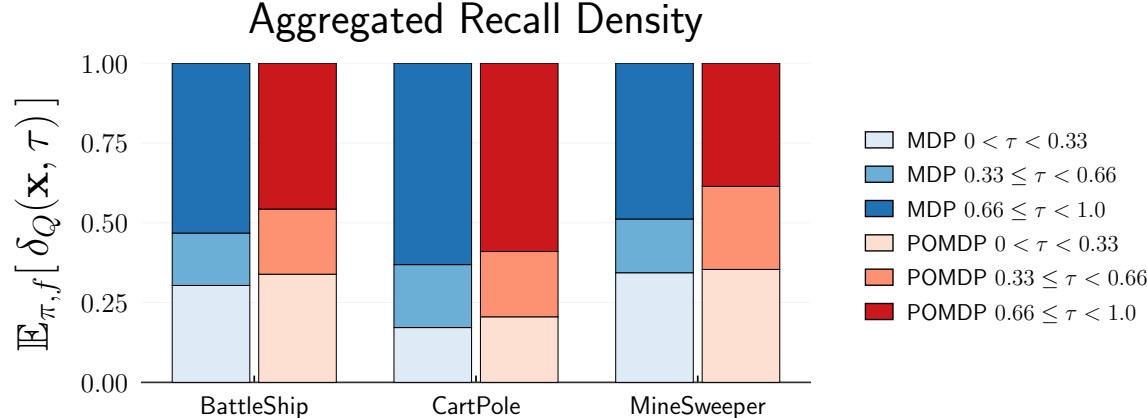

*Figure 22.* Aggregate Recall Density for DQN. We evaluate DQN on three representative Easy tasks. The results confirm that value smearing persists even with replay buffers.

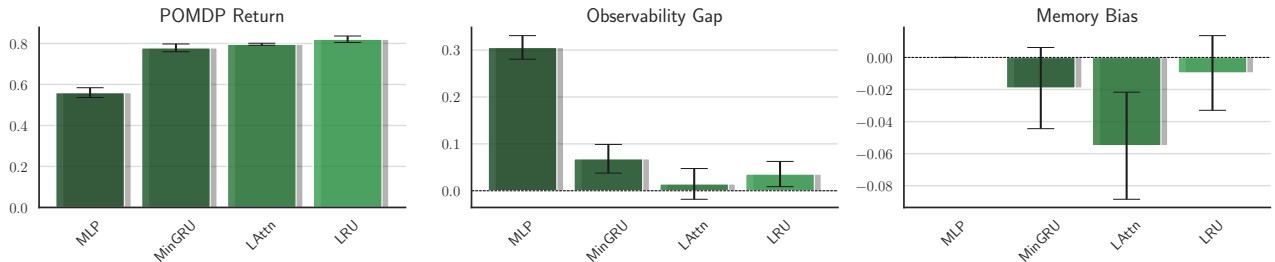

*Figure 23.* Disentangling returns for the DQN algorithm. We plot the aggregated POMDP returns, Observability Gap (Definition 4.1), and Memory Bias (Definition 4.2) across three environments and the Easy difficulty configuration. The results confirm that the negative memory bias and performance gaps persist in q-learning with replay buffer.

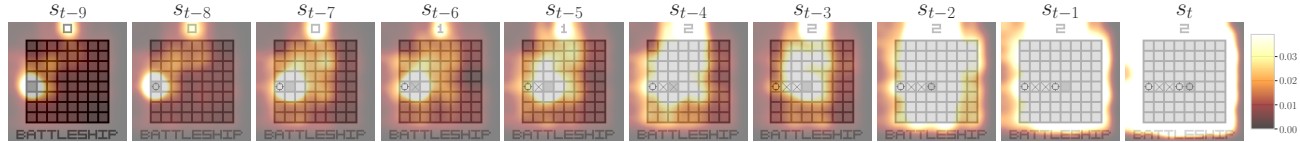

*Figure 24.* Pixel-level saliency for DQN with LRU on BattleShip MDP. We visualize the gradient of the Q-value with respect to past observations. Consistent with our PQN results, we observe value smearing even in the off-policy DQN setting.

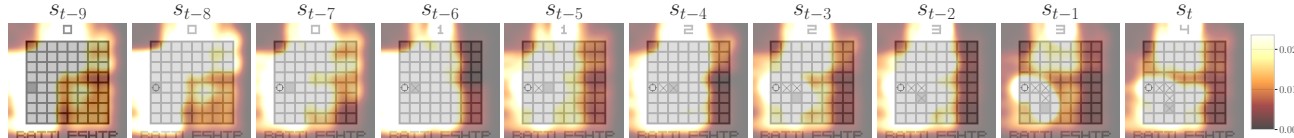

*Figure 25.* Pixel-level saliency for DQN with Linear Transformer on BattleShip MDP.

# D. Additional Recurrent State Contamination Experiments

We provide additional experiments for recurrent policy contamination. We use the cartpole example for even longer rollout horizons of 50 and 100 timesteps. We find that old observations still have a significant impact on the relative $Q$ values ($A$) and corresponding actions.

We also demonstrate this affect across different algorithms and memory models, such as PPO, DQN, and the Linear Transformer.

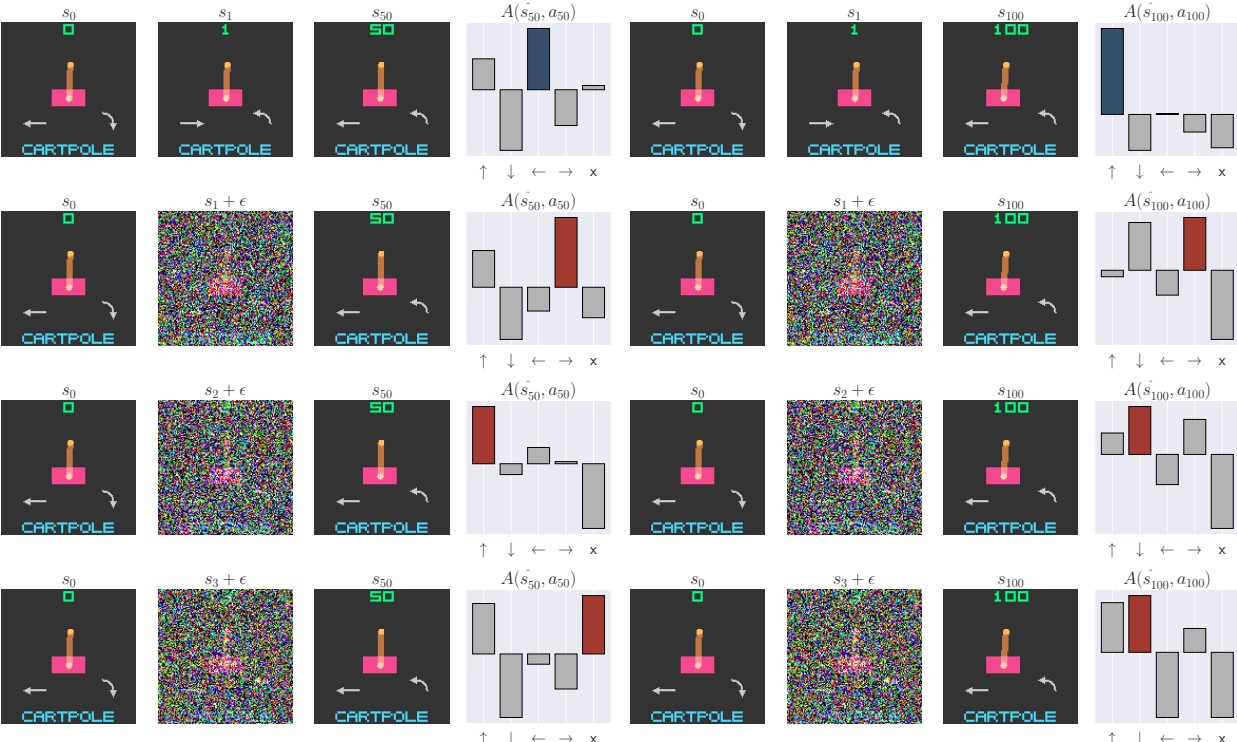

*Figure 26.* We demonstrate recurrent state contamination over longer horizons, using the LRU model. At 50 and 100 timesteps in the future, corrupted observations can still influence relative $Q$ values ($A$) enough to change the selected action.

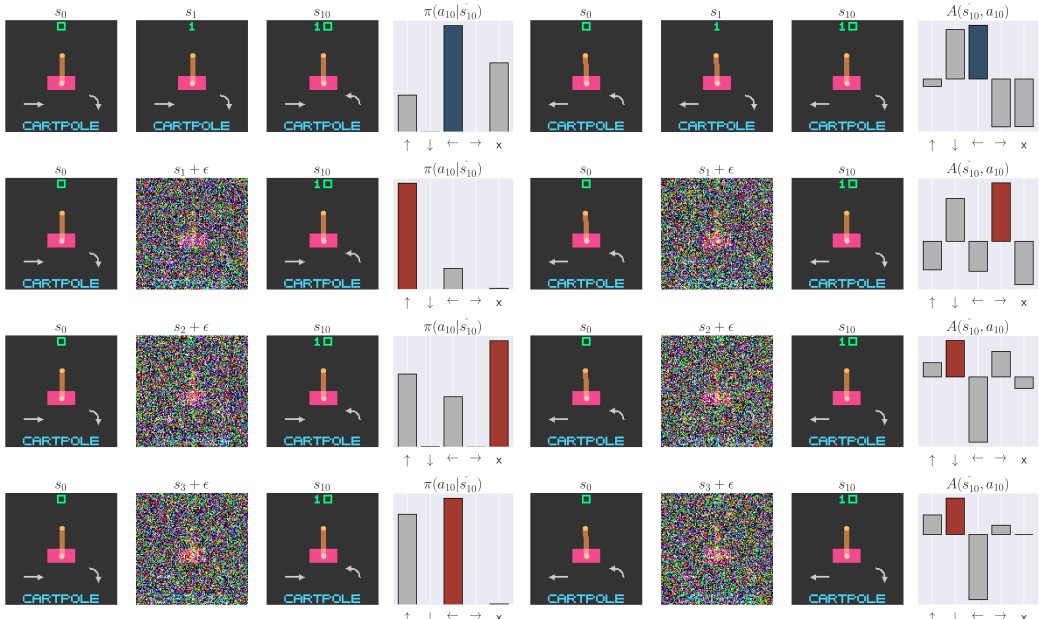

*Figure 27.* Noise injection experiments on additional baselines. **(Left) PPO with LRU**, we plot the policy distribution $\pi(a|s)$ after injecting noise, observing results similar to PQN. **(Right) DQN with Linear Transformer**, we illustrate the relative $Q$ values ($A$) after noise injection. Both experiments confirm that the vulnerability to state contamination persists across different methods and models.

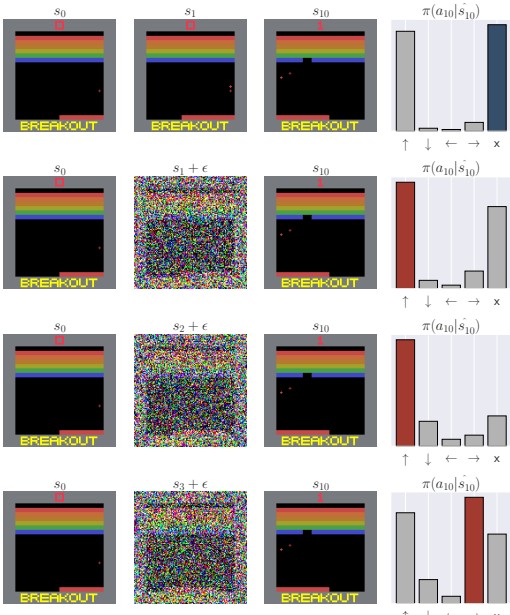

*Figure 28.* We demonstrate that injecting a noise frame early in an episode causes a change in the policy distribution $\pi(a|s)$ in PPO with GRU. The same phenomenon persists across different memory-based baselines and algorithms.

# E. Return Analysis by Model

In this section, we provide the unnormalized returns (mean and standard deviation) for the aggregated results in Fig. 5. We provide values for each environment, memory model, and difficulty level.

*Table 2.* Return for the LRU model, broken down by task and observability.

| Environment | MDP Return | POMDP Return |
|---|---|---|
| AutoEncodeEasy | 0.26 ± 0.00 | 0.27 ± 0.01 |
| AutoEncodeMedium | 0.24 ± 0.01 | 0.24 ± 0.00 |
| AutoEncodeHard | 0.23 ± 0.02 | 0.22 ± 0.02 |
| BattleShipEasy | 0.85 ± 0.02 | 0.84 ± 0.01 |
| BattleShipMedium | 0.78 ± 0.03 | 0.82 ± 0.01 |
| BattleShipHard | 0.73 ± 0.04 | 0.80 ± 0.01 |
| BreakoutEasy | 0.89 ± 0.01 | 0.75 ± 0.04 |
| BreakoutMedium | 0.83 ± 0.02 | 0.76 ± 0.06 |
| BreakoutHard | 0.84 ± 0.00 | 0.72 ± 0.09 |
| CartPoleEasy | 0.99 ± 0.00 | 0.99 ± 0.00 |
| CartPoleMedium | 0.98 ± 0.01 | 0.98 ± 0.01 |
| CartPoleHard | 0.97 ± 0.01 | 0.95 ± 0.02 |
| CountRecallEasy | 0.47 ± 0.06 | 0.36 ± 0.01 |
| CountRecallMedium | 0.22 ± 0.03 | 0.23 ± 0.04 |
| CountRecallHard | 0.15 ± 0.02 | 0.23 ± 0.01 |
| MineSweeperEasy | 0.87 ± 0.00 | 0.82 ± 0.01 |
| MineSweeperMedium | 0.67 ± 0.01 | 0.66 ± 0.01 |
| MineSweeperHard | 0.62 ± 0.01 | 0.61 ± 0.00 |
| NavigatorEasy | 0.91 ± 0.01 | 0.44 ± 0.09 |
| NavigatorMedium | 0.91 ± 0.01 | 0.32 ± 0.03 |
| NavigatorHard | 0.94 ± 0.01 | 0.40 ± 0.06 |
| NoisyPoleEasy | 0.99 ± 0.00 | 0.99 ± 0.00 |
| NoisyPoleMedium | 0.98 ± 0.01 | 0.99 ± 0.00 |
| NoisyPoleHard | 0.97 ± 0.01 | 0.97 ± 0.01 |
| SkittlesEasy | 0.83 ± 0.01 | 0.52 ± 0.02 |
| SkittlesMedium | 0.79 ± 0.01 | 0.51 ± 0.00 |
| SkittlesHard | 0.74 ± 0.01 | 0.47 ± 0.00 |
| TetrisEasy | 0.01 ± 0.00 | 0.01 ± 0.00 |
| TetrisMedium | 0.01 ± 0.00 | 0.01 ± 0.00 |
| TetrisHard | 0.01 ± 0.00 | 0.01 ± 0.00 |

*Table 3.* Return for the MinGRU model, broken down by task and observability.

| Environment | MDP Return | POMDP Return |
|---|---|---|
| AutoEncodeEasy | 0.27 ± 0.00 | 0.27 ± 0.00 |
| AutoEncodeMedium | 0.25 ± 0.01 | 0.26 ± 0.00 |
| AutoEncodeHard | 0.25 ± 0.00 | 0.25 ± 0.00 |
| BattleShipEasy | 0.82 ± 0.02 | 0.76 ± 0.01 |
| BattleShipMedium | 0.73 ± 0.02 | 0.75 ± 0.02 |
| BattleShipHard | 0.71 ± 0.03 | 0.73 ± 0.02 |
| BreakoutEasy | 0.88 ± 0.02 | 0.36 ± 0.02 |
| BreakoutMedium | 0.83 ± 0.01 | 0.33 ± 0.04 |
| BreakoutHard | 0.76 ± 0.03 | 0.28 ± 0.02 |
| CartPoleEasy | 0.98 ± 0.01 | 0.87 ± 0.02 |
| CartPoleMedium | 0.93 ± 0.02 | 0.68 ± 0.06 |
| CartPoleHard | 0.88 ± 0.08 | 0.54 ± 0.03 |
| CountRecallEasy | 0.31 ± 0.01 | 0.29 ± 0.03 |
| CountRecallMedium | 0.17 ± 0.00 | 0.21 ± 0.04 |
| CountRecallHard | 0.15 ± 0.01 | 0.13 ± 0.04 |
| MineSweeperEasy | 0.83 ± 0.01 | 0.76 ± 0.01 |
| MineSweeperMedium | 0.64 ± 0.00 | 0.61 ± 0.00 |
| MineSweeperHard | 0.58 ± 0.01 | 0.58 ± 0.01 |
| NavigatorEasy | 0.89 ± 0.02 | 0.33 ± 0.08 |
| NavigatorMedium | 0.74 ± 0.08 | 0.25 ± 0.12 |
| NavigatorHard | 0.13 ± 0.45 | 0.09 ± 0.08 |
| NoisyPoleEasy | 0.95 ± 0.01 | 0.89 ± 0.01 |
| NoisyPoleMedium | 0.92 ± 0.01 | 0.86 ± 0.01 |
| NoisyPoleHard | 0.86 ± 0.02 | 0.81 ± 0.01 |
| SkittlesEasy | 0.80 ± 0.00 | 0.39 ± 0.01 |
| SkittlesMedium | 0.76 ± 0.02 | 0.36 ± 0.01 |
| SkittlesHard | 0.73 ± 0.01 | 0.31 ± 0.01 |
| TetrisEasy | 0.01 ± 0.00 | 0.01 ± 0.00 |
| TetrisMedium | 0.01 ± 0.01 | 0.01 ± 0.01 |
| TetrisHard | 0.01 ± 0.00 | 0.00 ± 0.00 |

*Table 4.* Return for the GRU model, broken down by task and observability.

| Environment | MDP Return | POMDP Return |
|---|---|---|
| AutoEncodeEasy | $0.26 \pm 0.00$ | $0.26 \pm 0.00$ |
| AutoEncodeMedium | $0.23 \pm 0.01$ | $0.24 \pm 0.02$ |
| AutoEncodeHard | $0.23 \pm 0.01$ | $0.23 \pm 0.02$ |
| BattleShipEasy | $0.84 \pm 0.01$ | $0.84 \pm 0.01$ |
| BattleShipMedium | $0.76 \pm 0.05$ | $0.81 \pm 0.01$ |
| BattleShipHard | $0.71 \pm 0.04$ | $0.82 \pm 0.01$ |
| BreakoutEasy | $0.89 \pm 0.01$ | $0.62 \pm 0.10$ |
| BreakoutMedium | $0.85 \pm 0.01$ | $0.60 \pm 0.08$ |
| BreakoutHard | $0.85 \pm 0.02$ | $0.44 \pm 0.02$ |
| CartPoleEasy | $1.00 \pm 0.00$ | $0.98 \pm 0.01$ |
| CartPoleMedium | $0.99 \pm 0.01$ | $0.94 \pm 0.02$ |
| CartPoleHard | $0.96 \pm 0.01$ | $0.93 \pm 0.04$ |
| CountRecallEasy | $0.37 \pm 0.04$ | $0.36 \pm 0.01$ |
| CountRecallMedium | $0.19 \pm 0.02$ | $0.21 \pm 0.03$ |
| CountRecallHard | $0.16 \pm 0.04$ | $0.18 \pm 0.08$ |
| MineSweeperEasy | $0.85 \pm 0.01$ | $0.81 \pm 0.01$ |
| MineSweeperMedium | $0.66 \pm 0.00$ | $0.65 \pm 0.01$ |
| MineSweeperHard | $0.61 \pm 0.01$ | $0.61 \pm 0.00$ |
| NavigatorEasy | $0.90 \pm 0.01$ | $0.44 \pm 0.06$ |
| NavigatorMedium | $0.75 \pm 0.25$ | $0.34 \pm 0.09$ |
| NavigatorHard | $0.90 \pm 0.04$ | $0.38 \pm 0.06$ |
| NoisyCartPoleEasy | $0.98 \pm 0.01$ | $0.99 \pm 0.00$ |
| NoisyCartPoleMedium | $0.97 \pm 0.02$ | $0.98 \pm 0.01$ |
| NoisyCartPoleHard | $0.95 \pm 0.02$ | $0.97 \pm 0.02$ |
| SkittlesEasy | $0.86 \pm 0.01$ | $0.50 \pm 0.00$ |
| SkittlesMedium | $0.83 \pm 0.00$ | $0.45 \pm 0.00$ |
| SkittlesHard | $0.75 \pm 0.00$ | $0.40 \pm 0.00$ |
| TetrisEasy | $0.01 \pm 0.00$ | $0.01 \pm 0.00$ |
| TetrisMedium | $0.01 \pm 0.00$ | $0.01 \pm 0.00$ |
| TetrisHard | $0.01 \pm 0.00$ | $0.01 \pm 0.00$ |

*Table 5.* Return for the Linear Transformer model, broken down by task and observability.

| Environment | MDP Return | POMDP Return |
|---|---|---|
| AutoEncodeEasy | $0.25 \pm 0.00$ | $0.26 \pm 0.00$ |
| AutoEncodeMedium | $0.23 \pm 0.01$ | $0.23 \pm 0.01$ |
| AutoEncodeHard | $0.20 \pm 0.03$ | $0.21 \pm 0.02$ |
| BattleShipEasy | $0.66 \pm 0.09$ | $0.77 \pm 0.02$ |
| BattleShipMedium | $0.51 \pm 0.01$ | $0.52 \pm 0.02$ |
| BattleShipHard | $0.50 \pm 0.00$ | $0.50 \pm 0.00$ |
| BreakoutEasy | $0.10 \pm 0.04$ | $0.13 \pm 0.02$ |
| BreakoutMedium | $0.14 \pm 0.01$ | $0.11 \pm 0.05$ |
| BreakoutHard | $0.13 \pm 0.01$ | $0.10 \pm 0.05$ |
| CartPoleEasy | $0.98 \pm 0.01$ | $0.97 \pm 0.01$ |
| CartPoleMedium | $0.85 \pm 0.06$ | $0.78 \pm 0.05$ |
| CartPoleHard | $0.62 \pm 0.06$ | $0.60 \pm 0.04$ |
| CountRecallEasy | $0.39 \pm 0.03$ | $0.37 \pm 0.02$ |
| CountRecallMedium | $0.06 \pm 0.00$ | $0.10 \pm 0.04$ |
| CountRecallHard | $0.06 \pm 0.01$ | $0.05 \pm 0.00$ |
| MineSweeperEasy | $0.76 \pm 0.01$ | $0.73 \pm 0.04$ |
| MineSweeperMedium | $0.61 \pm 0.02$ | $0.59 \pm 0.02$ |
| MineSweeperHard | $0.55 \pm 0.00$ | $0.55 \pm 0.01$ |
| NavigatorEasy | $0.46 \pm 0.14$ | $0.36 \pm 0.05$ |
| NavigatorMedium | $-0.27 \pm 0.19$ | $-0.28 \pm 0.19$ |
| NavigatorHard | $-0.45 \pm 0.03$ | $-0.41 \pm 0.02$ |
| NoisyPoleEasy | $0.98 \pm 0.01$ | $0.97 \pm 0.02$ |
| NoisyPoleMedium | $0.96 \pm 0.02$ | $0.96 \pm 0.01$ |
| NoisyPoleHard | $0.92 \pm 0.02$ | $0.89 \pm 0.01$ |
| SkittlesEasy | $0.85 \pm 0.01$ | $0.32 \pm 0.02$ |
| SkittlesMedium | $0.80 \pm 0.01$ | $0.29 \pm 0.01$ |
| SkittlesHard | $0.75 \pm 0.00$ | $0.28 \pm 0.01$ |
| TetrisEasy | $0.01 \pm 0.00$ | $0.01 \pm 0.00$ |
| TetrisMedium | $0.01 \pm 0.01$ | $0.01 \pm 0.01$ |
| TetrisHard | $0.01 \pm 0.00$ | $0.00 \pm 0.00$ |

*Table 6.* Return for the Transformer model, broken down by task and observability.

*Table 7.* Return for the MLP model, broken down by task and observability.

| Environment | MDP Return | POMDP Return | Environment | MDP Return | POMDP Return |
|---|---|---|---|---|---|
| AutoEncodeEasy | $0.25 \pm 0.00$ | $0.26 \pm 0.00$ | AutoEncodeEasy | $0.26 \pm 0.00$ | $0.27 \pm 0.00$ |
| AutoEncodeMedium | $0.25 \pm 0.00$ | $0.25 \pm 0.00$ | AutoEncodeMedium | $0.25 \pm 0.00$ | $0.25 \pm 0.00$ |
| AutoEncodeHard | $0.25 \pm 0.00$ | $0.25 \pm 0.00$ | AutoEncodeHard | $0.25 \pm 0.00$ | $0.25 \pm 0.00$ |
| BattleShipEasy | $0.60 \pm 0.07$ | $0.56 \pm 0.03$ | BattleShipEasy | $0.85 \pm 0.02$ | $0.60 \pm 0.04$ |
| BattleShipMedium | $0.50 \pm 0.00$ | $0.50 \pm 0.00$ | BattleShipMedium | $0.70 \pm 0.04$ | $0.56 \pm 0.01$ |
| BattleShipHard | $0.50 \pm 0.00$ | $0.50 \pm 0.00$ | BattleShipHard | $0.75 \pm 0.03$ | $0.53 \pm 0.00$ |
| BreakoutEasy | $0.26 \pm 0.13$ | $0.21 \pm 0.07$ | BreakoutEasy | $0.93 \pm 0.01$ | $0.49 \pm 0.02$ |
| BreakoutMedium | $0.64 \pm 0.23$ | $0.18 \pm 0.03$ | BreakoutMedium | $0.89 \pm 0.02$ | $0.49 \pm 0.06$ |
| BreakoutHard | $0.34 \pm 0.27$ | $0.21 \pm 0.05$ | BreakoutHard | $0.84 \pm 0.02$ | $0.34 \pm 0.05$ |
| CartPoleEasy | $0.99 \pm 0.01$ | $0.97 \pm 0.01$ | CartPoleEasy | $0.99 \pm 0.00$ | $0.86 \pm 0.02$ |
| CartPoleMedium | $0.97 \pm 0.02$ | $0.90 \pm 0.03$ | CartPoleMedium | $0.96 \pm 0.01$ | $0.47 \pm 0.01$ |
| CartPoleHard | $0.92 \pm 0.04$ | $0.71 \pm 0.05$ | CartPoleHard | $0.95 \pm 0.01$ | $0.31 \pm 0.01$ |
| CountRecallEasy | $0.32 \pm 0.04$ | $0.10 \pm 0.06$ | CountRecallEasy | $0.40 \pm 0.04$ | $0.15 \pm 0.01$ |
| CountRecallMedium | $0.10 \pm 0.02$ | $0.04 \pm 0.00$ | CountRecallMedium | $0.17 \pm 0.02$ | $0.06 \pm 0.00$ |
| CountRecallHard | $0.07 \pm 0.01$ | $0.05 \pm 0.00$ | CountRecallHard | $0.15 \pm 0.01$ | $0.08 \pm 0.00$ |
| MineSweeperEasy | $0.80 \pm 0.01$ | $0.71 \pm 0.02$ | MineSweeperEasy | $0.86 \pm 0.01$ | $0.72 \pm 0.01$ |
| MineSweeperMedium | $0.62 \pm 0.03$ | $0.58 \pm 0.01$ | MineSweeperMedium | $0.65 \pm 0.01$ | $0.59 \pm 0.00$ |
| MineSweeperHard | $0.57 \pm 0.01$ | $0.55 \pm 0.00$ | MineSweeperHard | $0.57 \pm 0.00$ | $0.55 \pm 0.00$ |
| NavigatorEasy | $0.81 \pm 0.07$ | $0.12 \pm 0.21$ | NavigatorEasy | $0.88 \pm 0.02$ | $-0.20 \pm 0.05$ |
| NavigatorMedium | $-0.49 \pm 0.01$ | $-0.45 \pm 0.07$ | NavigatorMedium | $0.86 \pm 0.01$ | $-0.19 \pm 0.02$ |
| NavigatorHard | $-0.49 \pm 0.01$ | $-0.49 \pm 0.01$ | NavigatorHard | $0.94 \pm 0.01$ | $-0.07 \pm 0.08$ |
| NoisyPoleEasy | $0.98 \pm 0.01$ | $0.97 \pm 0.01$ | NoisyPoleEasy | $0.94 \pm 0.01$ | $0.82 \pm 0.02$ |
| NoisyPoleMedium | $0.96 \pm 0.01$ | $0.95 \pm 0.01$ | NoisyPoleMedium | $0.86 \pm 0.01$ | $0.74 \pm 0.01$ |
| NoisyPoleHard | $0.93 \pm 0.01$ | $0.90 \pm 0.02$ | NoisyPoleHard | $0.70 \pm 0.01$ | $0.66 \pm 0.01$ |
| SkittlesEasy | $0.81 \pm 0.02$ | $0.27 \pm 0.01$ | SkittlesEasy | $0.84 \pm 0.00$ | $0.41 \pm 0.00$ |
| SkittlesMedium | $0.77 \pm 0.01$ | $0.25 \pm 0.00$ | SkittlesMedium | $0.81 \pm 0.00$ | $0.37 \pm 0.00$ |
| SkittlesHard | $0.73 \pm 0.01$ | $0.23 \pm 0.00$ | SkittlesHard | $0.75 \pm 0.01$ | $0.33 \pm 0.00$ |
| TetrisEasy | $0.01 \pm 0.00$ | $0.01 \pm 0.00$ | TetrisEasy | $0.39 \pm 0.02$ | $0.39 \pm 0.02$ |
| TetrisMedium | $0.01 \pm 0.01$ | $0.01 \pm 0.01$ | TetrisMedium | $0.39 \pm 0.02$ | $0.39 \pm 0.02$ |
| TetrisHard | $0.01 \pm 0.00$ | $0.00 \pm 0.00$ | TetrisHard | $0.22 \pm 0.01$ | $0.22 \pm 0.01$ |

## F. Environment Descriptions

We describe each of our implemented MDPs and rules. Then, we explain how and why we make each MDP into a POMDP.

---

### Battleship

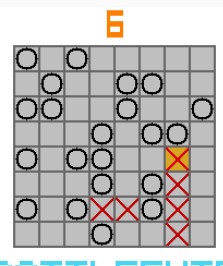

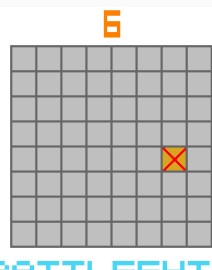

**MDP:** The agent's goal is to sink all the pre-placed ships to win the game. The agent controls a cursor that represents its current position on the grid. Each turn, the agent can either move the cursor one space or choose to "FIRE" the current grid cell. Each cell can be in one of three states: COVERED, HIT, or MISS. Initially, all cells are COVERED, but each changes to HIT or MISS upon firing. The agent receives a positive reward for hitting a ship, a neutral reward (0) for the first miss on a COVERED cell, and a negative reward for repeatedly firing HIT or MISS cells.

**POMDP:** We make this task partially observable by only showing the HIT or MISS markers for the tile the cursor currently occupies. All other tiles appear COVERED, regardless of whether they have been fired upon. This does not affect the reward function, only the observation function.

---

**Required Capabilities:** The agent must remember the tile label (HIT/MISS) of previously fired upon tiles, or risk running out of moves. This capability is critical for avoiding redundant actions, which incur negative rewards, and performing an efficient search of locations and ships. This will test spatial memory and integrate a sequence of localized observations into a coherent ability.

**MDP Recovery:** By retaining all prior observations, the agent can view all the same HIT/MISS markers as in the MDP.

---

### Count Recall

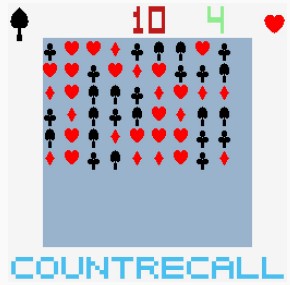

**MDP:** Every turn, the agent gets a value card and a query card. All previous value cards are displayed to the agent. The agent's task is to figure out how many times the query card has shown up so far. To do this, the agent counts how many times it has received the matching value card and uses that as the answer. If the agent guesses the correct count, it receives a positive reward. If the guess is incorrect, the agent receives no reward (0). This setup encourages the agent to accurately track and recall the frequency of specific cards over time.

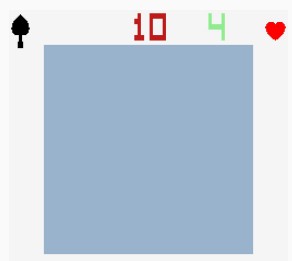

**POMDP:** All previous value cards are hidden, the agent may only observe the current value and query card.

---

**Required Capabilities:** The agent must learn to implement a latent counter for each card type to track the number of times a card has appeared. At each timestep, the agent must utilize these counters to answer the query.

**MDP Recovery:** By retaining all prior observations, the agent can reconstruct the MDP view from the suits at the top of the screen.

## Mine Sweeper

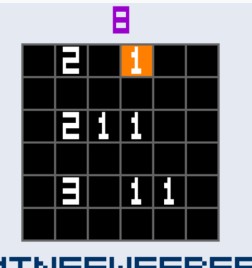

**MDP:** The agent's goal is to hit all the non-mine cells to win the game, inspired by the classic computer game Minesweeper. Similar to Battleship environments, the agent controls a cursor to navigate the grid and can either move to an adjacent cell or sweep the current cell each turn. Sweeping a safe cell earns the agent a positive reward and reveals the number of adjacent mines, while hitting a mine results in negative reward and ends the game. If the agent tries to sweep a cell that has already been revealed, it receives a small negative reward. This reward structure encourages the agent to explore efficiently, avoid mines, and minimize redundant actions to maximize its cumulative reward.

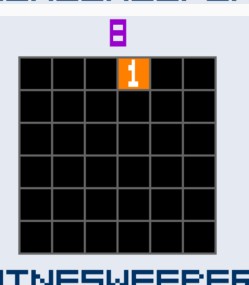

**POMDP:** In the POMDP variant, the agent can only observe the tile at the cursor position. All other tiles appear covered.

**Required Capabilities:** The agent must remember the number of adjacent mines for previously swept tiles, while simultaneously learning the fairly complex rules of the game. In particular, the agent must predict the location of mines from a partial view of adjacency information, which itself must be memorized.

**MDP Recovery:** By retaining all prior observations, the agent can view all the same tile values as in the MDP.

## Autoencode

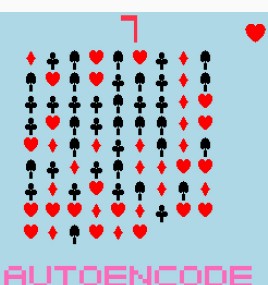

**MDP:** This game is similar to Simon but played in reverse. The agent's task is to recall and replay a sequence of cards in the opposite order they were shown. During the WATCH phase, the agent observes a randomly generated sequence of cards. This sequence is displayed on-screen. In the PLAY phase, the agent must reproduce the sequence in reverse order. The agent receives a positive reward for each correct card played and no reward (0) for incorrect choices.

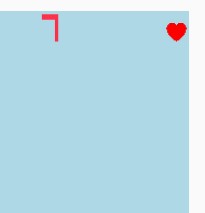

**POMDP:** This variant does not display the series of cards to the screen. At each timestep during the WATCH phase, the agent receives only the corresponding card.

**Required Capabilities:** By playing the cards in reverse order, the agent must learn to push and pop from a stack in latent space. Sequence permutation (pushing and popping) is well-known to be $NC^1$ complexity and not solvable by transformers in constant depth (Merrill et al., 2024). Only nonlinear RNNs can solve such a task in constant depth.

**MDP Recovery:** By retaining all prior observations, the agent can reconstruct the MDP view using the suits at the top of the screen.

## Navigator

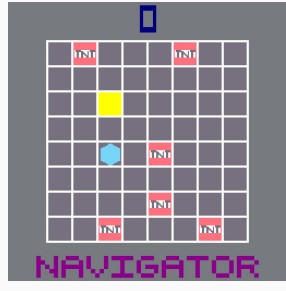

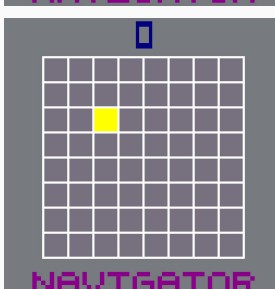

**MDP:** The agent's goal is to navigate and open the treasure on the board while avoiding the TNT. The agent navigates using a cursor that marks its current position. On each turn, it can either shift the cursor to an adjacent cell or decide to open the cell it is currently in. Every move the agent makes results in a small negative reward. If the agent lands on a TNT cell, it receives a significant negative reward, and the game ends immediately. This environment challenges the agent to discover the most efficient path to the treasure, aiming to maximize its cumulative reward by completing the game as quickly as possible.

**POMDP:** The agent only sees the position of the treasure and TNT blocks at the initial timestep. Afterwards, the agent can only see its current position.

**Required Capabilities:** The agent must remember the position of the treasure and TNT blocks over time. It must perform path planning within the learned memory state to reach the goal quickly.

**MDP Recovery:** By retaining the first observation, the agent has access to the goal location. Retaining all consecutive actions enables reconstruction of the agent's current location.

## Skittles

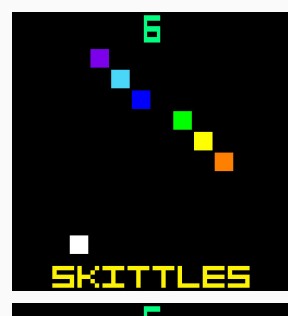

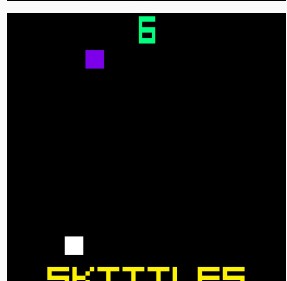

**MDP:** This game is inspired by many Atari games that focus on moving the main character to avoid touching enemies. The agent is a white square at the bottom of the screen and must dodge colored falling blocks. Upon touching a colored block, the game terminates. The agent receives a positive reward for surviving and a negative reward for touching a colored block. We spawn the colored blocks in such a way that the agent always has a feasible path for survival.

**POMDP:** In the POMDP variant, each colored block has only a 50% probability of being rendered at each timestep, causing them to flicker in and out of the agent's view, the agent cannot rely on the current observation to navigate safely.

**Required Capabilities:** The agent must use its memory to develop a form of object permanence, tracking the downward trajectories of blocks even when they are temporarily invisible. This requires the agent to maintain an internal belief about all colored blocks, integrating history observations to infer the complete observation of the environment.

**MDP Recovery:** The blocks have a 50% chance of rendering. Given that there are 11 tiles between the block spawn point and the agent, there is a $1 - 0.5^{11} = 0.9995$ chance that each block will be rendered at least once. By retaining all frames, the agent can reconstruct the position of all blocks (MDP state).

## Breakout

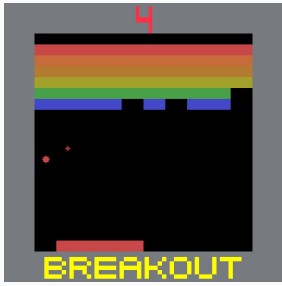

**MDP:** This game is based on the classic Atari Breakout game. The agent controls a paddle that they may use to deflect a moving ball. When the ball touches colored blocks, the blocks deflect the ball and then disappear. The agent receives positive reward when the ball collides with colored blocks, and a negative reward and termination condition when the ball passes the paddle and leaves the bottom of the play area. To make this game fully observable, we also provide a ball "tail" that determines the velocity direction and magnitude of the ball. Upon clearing all the blocks, the game terminates.

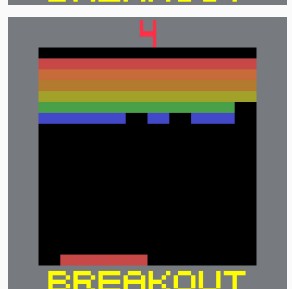

**POMDP:** In the POMDP variant, we remove the tail. Furthermore, the ball is only visible when moving upward. It becomes invisible when moving downward. We make sure to start the game with the ball moving upward so the agent knows the initial position and velocity of the ball.

**Required Capabilities:** Unlike other tasks, Breakout enables episodes that are thousands of timesteps long, allowing us to understand how memory adapts over very long sequences. This POMDP primarily tests short-term memory, as the agent need only (1) integrate position to predict velocity and (2) remember where the ball was a few timesteps ago to predict the future path of the ball. But it must learn a representation that can do so reliably over long durations.

**MDP Recovery:** By retaining all observations up to and including collision of the ball with the block, the agent can predict the deterministic angle and velocity at which the ball will reflect. This information is sufficient to predict the position and velocity of the ball, recovering the MDP.

## Tetris

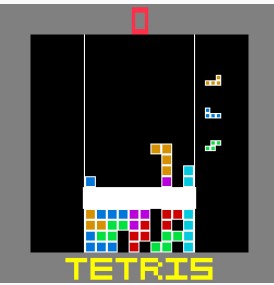

**MDP:** This is the well-known Russian puzzle game. Colored blocks of various shapes are dropped from the top of the screen, and the player controls their descent (position and rotation). Upon reaching the bottom of the screen or touching another block, the controlled block is frozen in-place and a new agent-controlled block is spawned. The blocks slowly pile up, and the agent receives a negative reward and termination upon the blocks reaching the top of the screen. Completely filling a row with blocks "clears" the row, deleting the tiles in the row and providing a positive reward. The game also terminates after clearing a predetermined number of rows.

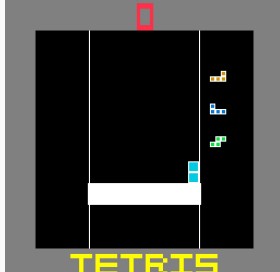

**POMDP:** In the POMDP variant, once a block is frozen in place, it becomes invisible. Only the currently controlled block and block-clear animation are visible to the agent. This is known as "In-visibile Tetris" and has a significant human following and some associated human competitions as well.

**Required Capabilities:** Invisible Tetris is arguably the most difficult task we propose. Standard Tetris is already a fairly hard task, as as far as we know, unsolved by RL. Even Tetris Grand Masters struggle with Invisible Tetris. The arrangement of tiles is complex and constantly shifting. It must be memorized perfectly, and a small error in state quickly compounds as the hidden structure undergoes mutation. This POMDP requires very strong state tracking capabilities and the ability to learn very difficult games.

**MDP Recovery:** Each block is visible during the timestep it is locked in place. Considering all observations therefore provides all block positions and recovers the MDP.

## CartPole

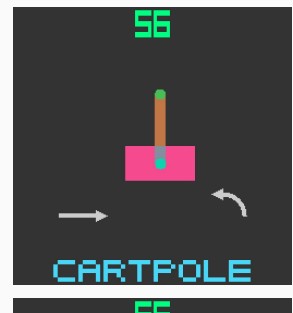

**MDP:** The CartPole environment is a classic control problem framed as a Markov Decision Process (MDP), where the objective is to balance a pole on a moving cart.

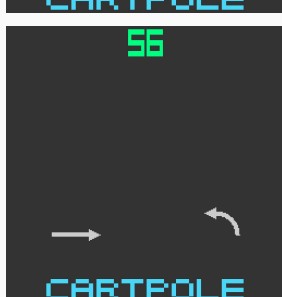

**POMDP:** In the POMDP version, after the initial timestep, the cart and pole are hidden from the agent while the arrows representing velocity remain. A horizontal arrow representing the cart's velocity and a curved arrow representing the pole's angular velocity. The size of each arrow is proportional to the magnitude of the corresponding velocity. This creates a pixel-form of the position-masked CartPole problem.

**Required Capabilities:** To succeed, the agent must integrate velocity signals from all prior timesteps to infer the hidden positional information, directly testing the model's capacity to maintain an internal state from a stream of partial information.

**MDP Recovery:** By considering the initial position (rendered) and integrating over all velocities, the agent can predict cart and pole position information.

## NoisyPole

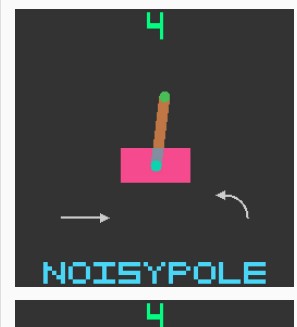

**MDP:** This task is Cartpole (Env. 1) affected by Gaussian noise on position and speed of the cart as well as angle and angular velocity of the pole. The result of noise is still reflected in the arrow magnitudes.

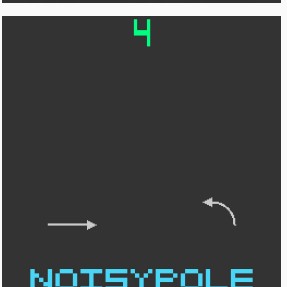

**POMDP:** After the initial timestep, the cart and pole are hidden from the agent, providing only the velocity arrows.

**Required Capabilities:** This is a harder version of the CartPole POMDP. (Morad et al., 2023a) demonstrates that certain memory-free policies can do well in POMDP CartPole, perhaps by learning a time-varying policy. Adding noise results in a better test of memory, as the agent must remember and react to the noise. A simple time-varying policy is insufficient to solve this task.

**MDP Recovery:** By considering the initial position (rendered) and integrating over all velocities, the agent can predict cart and pole position information. Noise information is reflected in the velocity observations, allowing integration to recover the full state.

# G. Hardware-Accelerated RL

Copying observations from CPU to GPU during rollouts is a major efficiency bottleneck in the training process (Lu et al., 2022). Using JAX, we can implement environments on accelerator hardware, avoiding copy overhead and training policies up to one thousand times faster (Lange, 2022; Koyamada et al., 2023; Matthews et al., 2024b). We highlight (Matthews et al., 2024a; Pignatelli et al., 2024; Lu et al., 2024) which offer hardware-accelerated POMDPs. With improved environment throughput, we can consider new training paradigms like Podracer (Hessel et al., 2021; Toledo, 2024). In this work, we focus on the PQN algorithm (Gallici et al., 2024), a simplified podracer version of Q learning. Unlike DQN (Mnih et al., 2015), PQN does not use target networks or replay buffers, and opts for an on-policy TD($\lambda$) objective.

Given the poor sample efficiency of RL, we focus on maintaining high environment throughput. First and foremost, we ensure that all environments are implemented in JAX (Bradbury et al., 2018) in a vectorizable and compilable form to leverage hardware acceleration. Beyond this, we find a number of small tricks can improve performance. Pre-caching sprites and their pixel positions upon reset provides noticeable performance gains, and replacing calls to `jax.lax.cond` with `jax.lax.switch` provides increased throughput.

## G.1. Vectorized State Transitions and Rendering

Our rendering system is built on a canvas framework. We draw to the canvas using primitives, such as letters, numbers, shapes, and more. All rendering functions are pure (without side effects), and may be easily vectorized or compiled. We build our rendering framework on top of `jax.numpy` and does not use any external dependencies. With these tools, anyone can easily render their own custom grid-based or card-based environments for further research and exploration. Seven of ten of our proposed environments come with fairly intricate rules, requiring multiple iterations of pattern rendering on the canvas. These complex tasks can be broadly split into two rendering categories: grid-based environments, like Battleship, and card-based environments, like CountRecall. Initially, we re-rendered at each step, but this was slow, even with JIT compilation. We found that removing dynamic computations improved performance, which we detail below.

**Grid-Based Environments**    In grid-based games, we divide the entire canvas into a bunch of small square cells, where each cell displays either the same or a unique pattern to reflect the environmental info contained in the current state. We precompute the coordinates of every cell that needs to be drawn on the canvas and stash them in two matrices – one for the x-axis positions and one for the y-axis. During rendering, we leverage `jax.vmap` to handle the process in parallel along the row dimension.

**Card-Based Environments**    We precompute all possible card templates (value, query, and historical cards) during initialization period, avoiding repetitive redrawing of static elements like suits or card positions. This turns dynamic rendering into a fast lookup-and-merge process. For example, when rendering value or query cards, we retrieve pre-drawn templates and apply them to the canvas using masks, skipping pixel-by-pixel logic during runtime. For the history display, we vectorize the drawing of historical cards using `jax.vmap`, generating all variations upfront and later selecting only the visible ones with fast array indexing. By JIT-compiling the render function, we lock these optimizations into a highly efficient, hardware-accelerated pipeline. The benefits are reduced per-frame computation, minimal branching, and GPU-friendly operations—all critical for real-time rendering. Even when rendering complex elements like the history grid, we avoid loops and instead use clever indexing with `jnp.argmax` and masks to overlay the latest valid symbols.

## H. Related Benchmarks

Below, we list additional relevant benchmarks and their characteristics. A tilde in the MDP Twins columns means that MDP and POMDP twins do not share observation spaces, making causal studies difficult. Our benchmark is the only one with POMDP/MDP twins and GPU acceleration.

| Benchmark | POMDP/MDP Twins | GPU |
|---|---|---|
| MuJoCo/MJX (Todorov et al., 2012) | | ✓ |
| Atari/ALE (Bellemare et al., 2013) | | |
| DMLab (Beattie et al., 2016) | | |
| MiniGrid/Navix (Chevalier-Boisvert et al., 2018), (Pignatelli et al., 2024) | ∼ | |
| Memory Task Suite (Fortunato et al., 2019) | | |
| MinAtar (Young & Tian, 2019) | | |
| EnvPool (Weng et al., 2022) | | ✓ |
| Gymnax (Lange, 2022) | | ✓ |
| POMDP Baselines (Ni et al., 2022) | ∼ | |
| MemoryGym (Pleines et al., 2022) | ✓ | |
| POPGym/POPJaxRL (Morad et al., 2023a), (Lu et al., 2024) | ∼ | |
| PGX (Koyamada et al., 2023) | | ✓ |
| MDP Playground (Rajan et al., 2023) | ✓ | |
| Jumanji (Bonnet et al., 2024) | | ✓ |
| Kinetix (Matthews et al., 2024b) | | ✓ |
| Craftax (Matthews et al., 2024a) | | ✓ |
| **POPGym Arcade (ours)** | ✓ | ✓ |

# I. Memory Models

In this appendix, we provide detailed descriptions of the memory models used in our experiments. We categorize these models into two families: classical recurrences, such as GRUs, which process sequences step-by-step, and associative recurrences, which leverage parallelizable operators for significantly faster computation.

## I.1. Classical Recurrences

Classical recurrent models, such as the Gated Recurrent Unit (GRU), process sequential information in a strictly ordered manner. The computation of the hidden state at any step $t$, denoted as $h_t = f(h_{t-1}, x_t)$, is fundamentally dependent on the completion of the previous step's computation, $h_{t-1}$. This creates a sequential chain of operations that cannot be parallelized.

**GRU**   The Gated Recurrent Unit (GRU) (Chung et al., 2014) simplifies the LSTM architecture by merging the cell state and hidden state into a single hidden state vector $h_t$. It uses two gates: a reset gate $r_t$ and an update gate $z_t$. The reset gate determines how to combine the new input with the previous hidden state, while the update gate decides how much of the previous hidden state to retain. The update equations are:

$$z_t = \sigma(W_z[h_{t-1}, x_t] + b_z) \tag{7}$$
$$r_t = \sigma(W_r[h_{t-1}, x_t] + b_r) \tag{8}$$
$$\tilde{h}_t = \tanh(W_h[r_t \odot h_{t-1}, x_t] + b_h) \tag{9}$$
$$h_t = (1 - z_t) \odot h_{t-1} + z_t \odot \tilde{h}_t. \tag{10}$$

The GRU's simpler structure makes it computationally more efficient than the LSTM while often achieving comparable performance.

## I.2. Associative Recurrences

Associative recurrent models, also known as linear recurrent models or memoroids (Morad et al., 2024), update the recurrent state with a binary operator • that obeys the associative property

$$h_3 = (x_1 \bullet x_2) \bullet x_3 = x_1 \bullet (x_2 \bullet x_3). \tag{11}$$

Given an associative recurrent update, we can leverage the associative property to rearrange the order of operations. For example, given four inputs $x_1, x_2, x_3, x_4$, we can compute either

$$h_4 = (((x_1 \bullet x_2) \bullet x_3) \bullet x_4) \qquad\qquad h_4 = (x_1 \bullet x_2) \bullet (x_3 \bullet x_4). \tag{12}$$

The former case corresponds to standard recurrent networks, relying on the prior recurrent state to compute the current state, resulting in linear time complexity. In the latter case, we can compute $x_1 \bullet x_2$ and $x_3 \bullet x_4$ in parallel, achieving logarithmic parallel time complexity (Hinze, 2004). (Blelloch, 1990) provide an associative scan implementation with linear space complexity. Associative recurrences are orders of magnitude faster than classical RNNs in practice, while using much less memory than transformers, making them useful in sample inefficient tasks like RL (Lu et al., 2024).

Our experiments rely on six distinct associative recurrent models: the traditional Transformer (Vaswani et al., 2017), the Fast Autoregressive Transformer (Linear Transformers) (Katharopoulos et al., 2020), a form of State-Space Model (Gu et al., 2022) called the Linear Recurrent Unit (LRU)(Orvieto et al., 2023), a GRU (Chung et al., 2014) variant of a Minimal Recurrent Network called the MinGRU (Feng et al., 2024), the Gated Delta Network (Yang et al., 2025), and the Test-Time Training (TTT) model (Sun et al., 2025). We provide formal descriptions of each model below.

**Attention**   We use self-attention from (Vaswani et al., 2017), replacing the original positional embedding with a RoPE embedding (Su et al., 2021). Attention can be written recurrently in the sense of updating a key-value cache, although a fixed cache size corresponds to a fixed window length. We compute attention via

$$h_t = \text{Softmax}\left(\frac{QK^\top}{\sqrt{d_k}}\right)V \qquad\qquad \hat{s}_t = h_t. \tag{13}$$

**Linear Transformers**   Standard transformers have quadratic space complexity from the outer product of keys and queries. The Fast Autoregressive Transformers (Linear Transformers) (Katharopoulos et al., 2020) replaces softmax attention with a kernelized attention mechanism to achieve linear space complexity and logarithmic time complexity via associative recurrent updates

$$h_t = h_{t-1} + \phi(W_k x_t)\phi(W_v x_t)^\top \qquad \hat{s}_t = \text{MLP}\left(x + \frac{\phi(W_q x_t)^\top h_t}{\phi(W_q x_t) \cdot \sum_{i=0}^t \phi(W_k x_i)}\right). \qquad (14)$$

Here, $\phi(x) = 1 + \text{ELU}(x)$ represents a kernel-space projection and the recurrent state $h$ represents the attention matrix. $W_k, W_v, W_q$ represent the key, query, and value projection parameters. To compute $\hat{s}_t$, we multiply the recurrent attention matrix by the query vector and normalize by a scalar.

**State-Space Models**   State-Space Models (SSMs) (Gu et al., 2022) model an associative recurrence via

$$h_t = \overline{W}_A h_{t-1} + \overline{W}_B x_t \qquad \hat{s}_t = \overline{W}_C h_t + \overline{W}_D x_t, \qquad (15)$$

where $\overline{W}_A, \overline{W}_B, \overline{W}_C, \overline{W}_D$ are discretizations of carefully initialized trainable parameters $W_A, W_B, W_C, W_D$. In practice, we initialize $W_A, W_B, W_C, W_D$ deterministically. Deterministic initialization applied to many consecutive SSM layers can result in instabilities, and so (Orvieto et al., 2023) proposes a meticulously derived random initialization and new parameterization of SSM parameters. They call their method the Linear Recurrent Unit (LRU).

**Minimal Recurrent Networks**   (Feng et al., 2024) revisit the popular Gated Recurrent Unit (GRU) (Chung et al., 2014) and Long Short-Term Memory (LSTM) (Hochreiter & Schmidhuber, 1997) RNNs. They simplify the GRU and LSTM recurrent updates, proposing the MinGRU and MinLSTM with efficient associative recurrent updates. The authors write the MinGRU as

$$h_t = (1 - \sigma(W_1 x_t + b_1)) \odot h_{t-1} + \sigma(W_1 x_t + b_1) \odot \tanh(W_2 x_t + b_2) \qquad \hat{s}_t = h_t, \qquad (16)$$

with trainable parameters $W_1, b_1, W_2, b_2$, sigmoid function $\sigma$, and elementwise product $\odot$. The authors find that the MinGRU outperforms multiple SSM variants across offline reinforcement learning tasks, using an offline decision transformer (Chen et al., 2021) framework.

**Gated Delta Networks**   (Yang et al., 2025) synthesize the fast, linear memory updates of Delta Networks (Yang et al., 2024) with the stable gating mechanics typical of modern recurrent models. the recurrent transition is formulated as an associative recurrence, where the update is expressed as

$$H_t = H_{t-1} + \sigma(W_\beta x_t + b_\beta)(W_v x_t - H_{t-1} W_k x_t)(W_k x_t)^\top \qquad \hat{s}_t = H_t(W_q x_t) \qquad (17)$$

where $W_\beta, b_\beta, W_k, W_v, W_q$ are learnable parameters, and $\sigma$ denotes the sigmoid activation function. The hidden state $H_t$ is a dynamic memory matrix updated via a gated variant of the Delta Rule (Schlag et al., 2021). The authors demonstrate that this gated update effectively expands the memory capacity of linear attention, the model achieve superior performance in long-context language modeling tasks.

**Test Time Training**   Test-Time Training (TTT) (Sun et al., 2025) introduce a novel paradigm redefining the recurrent hidden state as learnable weights. TTT Layers update their state with gradient descent steps on a self-supervised task at each time step even during inference. The sequence update is formulated as

$$W_t = W_{t-1} - \eta \nabla \mathcal{L}(W_{t-1}; x_t) \qquad \hat{s}_t = f(x_t; W_t) \qquad (18)$$

$W_t$ denotes the internal weights as the recurrent hidden state, $\mathcal{L}$ is the self-supervised reconstruction loss, $\eta$ is the learning rate, and $f$ represents the readout function. We optimize $W_t$ with closed form solution for single step gradient descent, and apply RoPE embedding (Su et al., 2021) with associative recurrence.

## J. Network Architecture

**PQN**  Our recurrent Q-network uses a hybrid CNN-RNN-MLP design. The RNN combines a 512-channel input tensor x with a 5-dimensional one-hot encoded last action vector from POPGym Arcade, processes these through its 512-unit hidden state, and generates 256-dimensional output features (Table 8).

*Table 8.* Recurrent Q network architecture

| Layer | Parameters | Activation |
|---|---|---|
| Conv2D | Channels: $3, 64$, Kernel: $5 \times 5$, Stride: $2$ | LeakyReLU |
| MaxPool2D | Kernel: $2 \times 2$, Stride: $2$ | – |
| Conv2D | Channels: $64, 128$, Kernel: $3 \times 3$, Stride: $2$ | LeakyReLU |
| MaxPool2D | Kernel: $2 \times 2$, Stride: $2$ | – |
| Conv2D | Channels: $128, 256$, Kernel: $3 \times 3$, Stride: $2$ | LeakyReLU |
| MaxPool2D | Kernel: $2 \times 2$, Stride: $2$ | – |
| Conv2D | Channels: $256, 512$, Kernel: $1 \times 1$, Stride: $1$ | LeakyReLU |
| RNN Cell | Input, Hidden, Output: $517, 512, 256$, Num Layer: $2$ | – |
| Linear | Features: $256 \rightarrow 256$ | LeakyReLU |
| LayerNorm | Dimensions: $256$ | – |
| **Output** | **Features: $256 \rightarrow 5$** | – |

**DQN**  Equivalent to the PQN architecture.

**PPO**  For PPO, we employ a fully decoupled architecture with distinct policy and value networks. Each network adopts a hybrid CNN-RNN-MLP structure identical to PQN, comprising a visual encoder that produces a 512-channel embedding followed by a two-layer recurrent block with 512 hidden dimension. The Output of RNN is subsequently mapped to a 5-dimensional categorical distribution for the actor and a scalar value estimate for the critic (Table 9).

*Table 9.* Recurrent Actor-Critic architecture

| Layer | Parameters | Activation |
|---|---|---|
| Conv2D | Channels: $3 \rightarrow 64$, Kernel: $5 \times 5$, Stride: $2$ | LeakyReLU |
| MaxPool | Kernel: $2 \times 2$, Stride: $2$ | – |
| Conv2D | Channels: $64 \rightarrow 128$, Kernel: $3 \times 3$, Stride: $2$ | LeakyReLU |
| MaxPool | Kernel: $2 \times 2$, Stride: $2$ | – |
| Conv2D | Channels: $128 \rightarrow 256$, Kernel: $3 \times 3$, Stride: $2$ | LeakyReLU |
| MaxPool | Kernel: $3 \times 3$, Stride: $1$ | – |
| Conv2D | Channels: $256 \rightarrow 512$, Kernel: $1 \times 1$, Stride: $1$ | LeakyReLU |
| RNN Cell | Input: $512$, Hidden: $512$, Output: $256$, Num Layer: $2$ | – |
| Linear | Features: $256 \rightarrow 256$ | LeakyReLU |
| LayerNorm | Dimensions: $256$ | – |
| **Actor Output** | **Features: $256 \rightarrow 5$** | – |
| **Critic Output** | **Features: $256 \rightarrow 1$** | – |

## K. Experiment Hyperparameters

We used one set of hyperparameters for all our PQN experiments. Please see the following section for the hyperparameter selection methodology.

*Table 10.* PQN hyperparameters used in all of our experiments. See (Gallici et al., 2024) for a detailed description of hyperparameters.

| Parameter | Value |
| --- | --- |
| TOTAL_TIMESTEPS | 10e6, 20e6 |
| TOTAL_TIMESTEPS_DECAY | 1e6, 2e6 |
| NUM_ENVS | 16 |
| NUM_STEPS | 128 |
| NUM_MINIBATCHES | 16 |
| NUM_EPOCHS | 4 |
| EPS_START | 1.0 |
| EPS_FINISH | 0.05 |
| EPS_DECAY | 0.25 |
| NORM_INPUT | False |
| NORM_TYPE | layer norm |
| LR | 0.00005 |
| MAX_GRAD_NORM | 0.5 |
| LR_LINEAR_DECAY | True |
| REW_SCALE | 1.0 |
| GAMMA | 0.99 |
| LAMBDA | 0.95 |

*Table 11.* PPO hyperparameters used in our experiments.

| Parameter | Value |
| --- | --- |
| TOTAL_TIMESTEPS | 10e6, 20e6 |
| TOTAL_TIMESTEPS_DECAY | 1e6, 2e6 |
| NUM_ENVS | 16 |
| NUM_STEPS | 128 |
| NUM_MINIBATCHES | 16 |
| UPDATE_EPOCHS | 4 |
| GAMMA | 0.99 |
| GAE_LAMBDA | 0.95 |
| CLIP_EPS | 0.2 |
| ENT_COEF | 0.01 |
| VF_COEF | 0.5 |
| NORM_TYPE | layer norm |
| LR | 0.0001 |
| MAX_GRAD_NORM | 0.5 |
| LR_LINEAR_DECAY | True |

## L. Hyperparameter Selection Methodology

To select hyperparameters, we performed a manual sweep over four model architectures (MLP, MinGRU, LRU, and Linear Transformers) across six tasks (CartPole, Navigator, BattleShip, MineSweeper, CountRecall, and AutoEncode), and all three difficulty levels. We evaluated the training timesteps, learning rate and schedule, exploration parameters, batch size, number of minibatches per epoch, gradient clipping magnitude. These refer to TOTAL_TIMESTEPS, LR, LR_LINEAR_DECAY, TOTAL_TIMESTEPS_DECAY, EPS_START, EPS_FINISH, EPS_DECAY, NUM_STEPS, NUM_ENVS, NUM_MINIBATCHES, MAX_GRAD_NORM respectively.

Unlike the original PQN paper which annealed learning epsilon to near-zero over the entire training duration, we found it beneficial to decay more quickly to a slightly higher final epsilon value. When experimenting with epsilon decay to 0.01 and 0.05 over the episode, we noticed that most learning tended to happen near the ends of training. By quickly annealing to 0.05, the policies learned much more quickly. We found annealing epsilon to 0.01 produced suboptimal results.

The batch size is a function of the number of workers and the number of steps taken at each epoch. We found decreasing the number of steps below 128 hurt performance, and increasing it beyond 128 yielded similar results but reduced sample efficiency. With a number of steps at 128, we found that doubling the batch size via double the number of workers, did not produce a noticeable improvement. Performance increased as we increased the number of minibatches to 16, beyond which we did not see meaningful improvements.

We found that changing lambda from the PQN recommended 0.65 to our selected 0.95 resulted in the best returns. We evaluated learning rates including 0.0005, 0.0001, 0.00005, 0.00001 finally settling on 0.00005. We found linear LR decay outperformed no decay, and that decaying over the first tenth of an episode produced better results than decaying over the full training duration. We tested gradient clipping values of 1.0 and 0.5, selecting the smaller value for training stability despite a minor efficiency reduction.

# M. Human Baselines

We added human baseline results in our study. Each human was provided the docstring associated with each game, then played the game using the arrow keys and spacebar. The scores were recorded and uploaded for our analysis.

These baselines are computed from five participants, matched against the five seeds reported in the Fig. 5. We report the per-task breakdown of normalized returns in $[0, 1]$ at the end of this response.

In general, humans tend to perform better on card games, perform similarly on board games, and perform worse on control tasks. Overall, the MLP outperforms humans on MDPs. Humans make up this gap when introducing partial observability, scoring almost the same as the MLP.

*Table 12.* Return for the human baselines, broken down by task and observability.

| Environment | MDP Return | POMDP Return |
|---|---|---|
| AutoEncodeEasy | $0.79 \pm 0.39$ | $0.28 \pm 0.09$ |
| AutoEncodeMedium | $0.82 \pm 0.17$ | $0.27 \pm 0.08$ |
| AutoEncodeHard | $0.58 \pm 0.12$ | $0.28 \pm 0.04$ |
| BattleShipEasy | $0.96 \pm 0.09$ | $0.71 \pm 0.08$ |
| BattleShipMedium | $0.92 \pm 0.08$ | $0.73 \pm 0.29$ |
| BattleShipHard | $0.94 \pm 0.13$ | $0.66 \pm 0.22$ |
| CartPoleEasy | $0.22 \pm 0.11$ | $0.18 \pm 0.10$ |
| CartPoleMedium | $0.16 \pm 0.10$ | $0.07 \pm 0.05$ |
| CartPoleHard | $0.12 \pm 0.12$ | $0.07 \pm 0.05$ |
| CountRecallEasy | $0.78 \pm 0.10$ | $0.35 \pm 0.20$ |
| CountRecallMedium | $0.68 \pm 0.17$ | $0.15 \pm 0.08$ |
| CountRecallHard | $0.57 \pm 0.11$ | $0.14 \pm 0.09$ |
| MineSweeperEasy | $0.78 \pm 0.16$ | $0.79 \pm 0.13$ |
| MineSweeperMedium | $0.41 \pm 0.21$ | $0.30 \pm 0.14$ |
| MineSweeperHard | $0.26 \pm 0.28$ | $0.17 \pm 0.17$ |
| NavigatorEasy | $0.97 \pm 0.02$ | $0.97 \pm 0.02$ |
| NavigatorMedium | $0.97 \pm 0.01$ | $0.97 \pm 0.01$ |
| NavigatorHard | $0.97 \pm 0.01$ | $0.68 \pm 0.67$ |
| NoisyPoleEasy | $0.09 \pm 0.05$ | $0.16 \pm 0.12$ |
| NoisyPoleMedium | $0.19 \pm 0.13$ | $0.17 \pm 0.08$ |
| NoisyPoleHard | $0.18 \pm 0.09$ | $0.14 \pm 0.12$ |
| BreakoutEasy | $0.55 \pm 0.45$ | $0.08 \pm 0.03$ |
| BreakoutMedium | $0.81 \pm 0.39$ | $0.06 \pm 0.02$ |
| BreakoutHard | $0.72 \pm 0.39$ | $0.06 \pm 0.05$ |
| SkittlesEasy | $0.80 \pm 0.33$ | $0.54 \pm 0.33$ |
| SkittlesMedium | $0.76 \pm 0.32$ | $0.59 \pm 0.39$ |
| SkittlesHard | $0.69 \pm 0.30$ | $0.42 \pm 0.24$ |
| TetrisEasy | $0.21 \pm 0.14$ | $0.02 \pm 0.03$ |
| TetrisMedium | $0.18 \pm 0.12$ | $0.03 \pm 0.03$ |
| TetrisHard | $0.03 \pm 0.04$ | $0.00 \pm 0.00$ |

## N. LLM Usage

We used LLMs as a writing aid for portions of the paper. We used it to improve grammar and clarity, as well as to restructure the order in which we present our ideas.

