# OpenReview forum: "Investigating Memory in Model-Free RL with POPGym Arcade"
_ICML.cc/2026/Conference — ICML 2026 spotlight_

### Official Review · Reviewer_FW14 · 2026-03-11

**Soundness:** 3
**Presentation:** 3
**Significance:** 3
**Originality:** 2
**Overall Recommendation:** 5
**Confidence:** 4

**Summary:**

This paper proposes POPGym Arcade as a new benchmark for evaluating DeepRL memory models on Atari style arcade games. The benchmark differentiates itself by providing paired MDP and POMDP versions of each environment to enable controlled studies that disentangle different aspects of the models performance that are often not made clear by return alone. Specifically, the authors provide analysis of the observability gap, memory bias, and recall density while also providing visualizations of the impact of pixels on memory. The authors through experiments then demonstrate that memory introduces architectural bias and that models tend to incorrectly assign value to irrelevant history.  The authors also provide experiments showing how brittle policies are to irrelevant past inputs.

**Compliance With Llm Reviewing Policy:**

Affirmed.

**Key Questions For Authors:**

Q1: What interventions can potentially be done to impact the memory bias / observability gap tradeoff that warrants separating them out as separate metrics?

Q2: For your metrics based on the Q function do you use the true Q function or the agent's own approximation of the Q function?

Q3:  For a new arbitrary memory architecture, how do you use your codebase to leverage the four key memory inspection metrics / tools highlighted in your contributions?

Q4: In your experiments, how do you reconcile the lack of convergence with the key insights that you highlight? Wouldn't it be unexpected if your results turned out differently?

**Limitations:**

Yes. This seems out of scope for this paper.

**Strengths And Weaknesses:**

**Overall:** I really like the core idea of this paper to leverage the dual POMDP and MDP version of a problem for deeper analysis of memory models for Deep RL. I also think it is great that the authors have been able to demonstrate impressive throughput to enable fast experimentation. The presentation of this paper is generally pretty good and the experiments seem sound with anonymized code provided. The originality of various aspects of this paper is more of a weak point -- although this is often the case for benchmark papers. I believe this is definitely not the first paper to consider paired MDP and POMDP versions of the same problem, but I am not aware of another memory benchmarks that highlights this point so front and center.

**Tools to Measure and Interpret Memory:** I struggle to really gauge the level of contribution of this part. The analysis leading to these definitions is very simple and also is very much defined by construction rather than presented from first principles. In particular, the way that the Q function is presented bugs me. It is presented as if it is the true Q function, but I believe it is the estimated Q function by the current model to make this possible to measure in practice? I wonder to what degree this undermines the credibility of these metrics. Additionally, while the disentangling of the observability gap and memory bias is interesting, I am not sure what can really be actionable when doing this. It seems like a tradeoff inherent to these different architectures, so I am not sure what interventions can be suggested by it.

**Code:** The key contribution of this benchmark paper would obviously be the codebase provided by the authors and the ease by which it allowed others to run experiments testing out different memory models. The README highlights "Memory Introspection Tools" but it is not clear how to easily get results for the other key metrics highlighted in the paper. On the surface this seems non-trivial to me for a new arbitrary memory architecture because it involves using the same architecture to process different kinds of input. If it still involves significant work to i.e. get measurements of the observability gap and memory bias for a new architecture, that would seem to greatly limit the potential impact of the codebase.

**Empirical Results:** I find the results provided to be of somewhat limited novelty, which may undermine the potential motivation of future researchers to test out their model in this benchmark. The fact that memory introduces architectural bias is obvious by construction. It is interesting that this bias can be large, but it isn't that surprising to me as the policies are not converged. Again, the fact that memory models incorrectly assign value to irrelevant history seems like it obviously must be the case when the policies have not yet converged. Finally, the recurrent state contamination results are also not surprising in light of the entire literature on adversarial examples and OOD generalization. It additionally highlights the deficiency of the training paradigm of this benchmark which does not train on variations such as those introduced in the training of foundation models or in benchmarks like ProcGen. That said, it is nice to provide a tool so that new architectures can easily be analyzed in this setting.

---

> ### Author Rebuttal · Authors · 2026-03-31
>
> Thank you for taking the time to read our paper, we really appreciate the thoroughness of your review.
>
> > Q1: What interventions can potentially be done to impact the memory bias / observability gap...
>
> Observability Gap and Memory Bias roughly form a Pareto front for memory model complexity.
>
> Large observability gap $\implies$ Memory model lacks capacity to recover Markov states, results in state aliasing
> - Intervention 1: Increase model capacity (deeper RNN, larger recurrent state)
> - Intervention 2: Ensure BPTT gradient does not vanish over trajectory
>
> Large negative memory bias $\implies$ Introducing memory harms training, loss landscape too difficult for optimizer
> - Intervention 1: Reduce model complexity (fewer layers, smaller recurrent size)
> - Intervention 2: Regularization on memory model (skip connections, normalization)
>
> We peformed an RNN depth ablation to demonstrate this tradeoff. Deeper RNNs decrease the Gap but produce larger negative Bias.
>
> |Model/Task|RNN Depth|Obs. Gap $\downarrow$|Memory Bias $\uparrow$|
> |--|--|--|--|
> MinGRU/BattleShip|1|0.0394|-0.0049
> ||8|0.0005|-0.3526
> LRU/Minesweeper|1|0.0727|+0.0019
> ||8|0.0378|-0.0039
> Attention/MineSweeper|1|0.1044|-0.0432
> ||8|0.1018|-0.0992
>
> We will also ablate recurrent size, and other depth/model/task configurations and seeds in the final paper.
>
> > Q2: ...do you use the true Q function or the agent's own approximation of the Q function?
>
> We want to understand what the memory model is learning, so we plot the learned $Q$ function in our gradient and Recall Density plots. We stress our explanation for this in Section 6.3:
>
> >> We highlight the MDP results because MDPs have a known ground-truth credit distribution: the return depends only on the current state/observation.
>
> Computing $Q_*$ analytically for the POMDPs is intractable, but we know that $Q_*$ for MDPs **relies only on the current observation.** So the **MDP** analyses in Fig. 7, 8, 9 demonstrate that memory is not learning a compact, one-observation representation as one might expect. Note **we replicate all experiments on both MDPs and POMDPs** (see Appendix), but without ground truth $Q_*$, POMDP results are difficult to interpret.
>
> > Q3: For a new arbitrary memory architecture, how do you use your codebase to leverage the four key memory inspection metrics...
>
> The code/branch you have faithfully reproduces our exact results, but relies on an older version of `jax`. We have an updated public repository that provides 17 existing memory models. For a new a memory model, you must implement a memory model then register it, we will register it as `my_rnn` in this example. Your model must expose two methods:
> - `__call__(self, recurrent_state, (obs, done)) -> recurrent_state, markov_state`
> - `initialize_carry(self) -> recurrent_state`
>
> Reproduce all plots for `my_rnn` via:
> ```bash
> # Train, upload metrics, output model weights
> python train.py --MEMORY_TYPE=my_rnn --PROJECT=my_rnn
> # Download run CSV from wandb then plot gap/bias
> python plotting/download_csv.py \
>     --entity wandb_entity \
>     --project=wandb_project_name \
>     --model-group-csv my_rnn.csv
> python plotting/return_gap_bias.py \
>     --input-csv my_rnn.csv \
>     --output gap_bias.pdf
> # Plot pixel saliency
> python plotting/pixel_vis_pqn \
>     --model-path my_rnn_weight.pkl \
>     --env-name CartPoleEasy \
>     --memory_type my_rnn \
>     --output pixels.pdf
> # Compute recall density as CSV, then plot it
> python plotting/density_analysis_pqn.py \
>     --model-dir model_weights_dir \
>     --out_dir your_recall_density_dir
> python plotting/plot_density_summary.py \
>     --recall_density_dir your_recall_density_dir \
>     --output density.pdf
> # Noise injection plots
> python plotting/noiseva.py \
>     --model-dir /path/to/checkpoints \
>     --memory-types my_rnn \
>     --env-names CartPoleEasy
> ```
>
> > Q4: ...how do you reconcile the lack of convergence...
>
> If all tasks were already solved, the benchmark would not be useful. You are right that investigating non-converged models is misleading. We discuss this in the Fig. 6 caption:
>
> >> Particular models [and tasks] that we focus [our investigation] on (LRU, GRU) appear converged with low variance across seeds, ruling out optimization instability as a confounder for the value smearing experiments.
>
> Note that the main paper experiments focus on **MDPs** which are virtually all converged for LRU/GRU in Fig 6.
>
> We also replotted Fig. 6 after training for 100M env steps (10x longer) and the results are nearly identical. We lack compute to run all experiments/seeds for 10x longer, but we can run a specific experiment or model/task configuration you are interested in for 10x longer.
>
> > ...this benchmark which does not train on variations such as those introduced in the training of foundation models or in benchmarks like ProcGen.
>
> All tasks use randomized initial states. Tile RGB values are part of each task configuration and easy to change. If accepted, we will add an option to randomize tile colors on `reset`.

---

> > ### Author Rebuttal · Reviewer_FW14 · 2026-04-01
> >
> > Thank you for the detailed response to my review. I feel that the authors have provided satisfactory answers to my main questions and concerns that really got at the core aspects of my criticisms. I will update my score accordingly.

---

### Official Review · Reviewer_bM8f · 2026-03-11

**Soundness:** 4
**Presentation:** 3
**Significance:** 3
**Originality:** 4
**Overall Recommendation:** 5
**Confidence:** 4

**Summary:**

This paper provides a deeper investigation in how memory models are evaluated in deep RL. The authors argue that comparing the memory capabilities of diffferent models is not straightforward as it faces confounding factors like the compatibility of the base algorithm and the memory module, optimization issues and architectural biases.


To address this, the authors introduce POPGym Arcade, a highly parallelizable, JAX-based suite of environments that provides paired MDP (fully observable) and POMDP (partially observable) variants of the exact same tasks. Using these environment twins, they introduce two new evaluation metrics: the observation gap and the memory bias. Furthermore, they develop gradient-based diagnostic tools, such as recall density, to explicitly measure which past observations the agent relies on to make current decisions.


Applying these tools, the authors uncover a widespread failure mode in RL memory architectures they term "value smearing." They find that memory modules fail to assign proper credit to past events because of overly relying on previous observations, even when the entire state information is included in the current observation.

**Compliance With Llm Reviewing Policy:**

Affirmed.

**Final Justification:**

All of my questions were answered during the rebuttal. I maintain my positive score.

**Key Questions For Authors:**

1) Key Questions For Authors: Have you experimented with any regularization techniques (such as information bottlenecks or explicit state-reconstruction auxiliary losses) to encourage the memory model to drop irrelevant history and mitigate value smearing?
2) The main text relies heavily on PQN to isolate value function approximation. Does the magnitude of the Observability Gap or Memory Bias differ significantly when using experience replay / target networks?

**Limitations:**

yes

**Strengths And Weaknesses:**

Strengths:

1) While the RL community has recognized the importance of memory in partially observable settings, a deeper invesstigation into the existence of confounding factors was missing. This paper does exactly this and provides measurble metrics like the observability gap and memory bias.
2) I think this is a timely paper as it proposes ways to ensure proper empirical practises. The benchmark environments are simple and fast, hence are excellent diagnostic tools.
3) Creating an environment suite where the fully observable and partially observable twins share an identical, high-dimensional pixel observation space and action space is a is important for performing counterfactual studies.
4) The authors support their claims by testing across five distinct memory architectures (Transformer, Linear Transformer, MinGRU, GRU, LRU) and demonstrating that the findings generalize across fundamentally different RL algorithms.
5) The environments are implemented purely in jax, allowing for massive parallelization (10,000 times faster throughput than standard atari simulators).

Weakness:

1) The transition from defining the conceptual tools into the gradient-based Recall Density equations is quite dense. The derivation could benefit from more intuitive, plain-text explanations.

---

> ### Author Rebuttal · Authors · 2026-03-29
>
> We sincerely thank the reviewer for this constructive feedback and recognition of our work. We will address each point below.
>
> >The transition from defining the conceptual tools into the gradient-based Recall Density equations is quite dense. The derivation could benefit from more intuitive, plain-text explanations.
>
> We have rewritten the Recall Density subsection to focus on what it measures, rather than the steps we use construct it. It's meaning should be more clear to readers now. Most construction details have been moved to the Appendix.
>
> > Have you experimented with any regularization techniques (such as information bottlenecks or explicit state-reconstruction auxiliary losses) to encourage the memory model to drop irrelevant history and mitigate value smearing?
>
> This is an insightful point. Honestly, our current results suggest that regularization can be part of the solution, but it is insufficient to fully resolve the issue. We are currently working on something to tackle this, but it still requires further research.
>
> >Does the magnitude of the Observability Gap or Memory Bias differ significantly when using experience replay / target networks?
>
> Our DQN results are similar to the PQN results in that we see similar Recall Density and smearing phenomena (Appendix C). Additionally, we started new experiments for the DQN Observability Gap and Memory Bias. See the following table for preliminary results on **a subset of easy tasks**. Scores will be lower once medium/hard tasks complete, and will further change as we add random seeds and more tasks.
>
> | Memory Model | POMDP Return | Observability Gap | Memory Bias |
> | :--- | :--- | :--- | :--- |
> | MinGRU | 0.76 | 0.14 | 0.05 |
> | LAttn | 0.82 | 0.04 | 0.01 |
> | LRU | 0.87 | -0.01 | 0.01 |
>
> Thank you again for taking the time to read our work. Please let us know if you have any more concerns.

---

> > ### Author Rebuttal · Reviewer_bM8f · 2026-04-01
> >
> > Thank you for your answers. All of my questions were answered. I maintain my positive score.

---

### Official Review · Reviewer_9qUy · 2026-03-13

**Soundness:** 2
**Presentation:** 3
**Significance:** 2
**Originality:** 2
**Overall Recommendation:** 4
**Confidence:** 4

**Summary:**

The paper tackles how to analyze and understand memory mechanisms in partially observable environments. To address limitations of existing evaluation practices, the paper introduces POPGym Arcade, a hardware-accelerated benchmark consisting of Atari-style environments with paired MDP/POMDP twins that share the same observation and action spaces. These paired environments enable controlled experiments to isolate the effect of memory on performance.

In addition to the benchmark, the authors propose several memory analysis tools, including the *Observability Gap* to quantify the difficulty of reconstructing state from observations, the *Memory Bias* to measure performance changes introduced by memory architectures, and visualization-based tools such as pixel-level gradient analysis and *Recall Density* to study how past observations influence current decisions.

Using these tools, the paper performs a systematic empirical study of several memory architectures (e.g., GRU, transformer-based models, and state-space models). The analysis reveals a previously underexplored phenomenon called value smearing, where value estimates incorrectly attribute credit to irrelevant past observations. The authors further demonstrate that this pathology can lead to recurrent state contamination, where out-of-distribution observations propagate through memory and influence policies far into the future.

**Compliance With Llm Reviewing Policy:**

Affirmed.

**Final Justification:**

**Revised Recommendation**

After the rebuttal, I still believe that the paper’s scope is narrower than the broader problem of memory in reinforcement learning. Memory in RL is closely connected to representation learning, and model-based approaches are an important paradigm for handling partial observability through latent-state modeling. From that perspective, restricting the paper to policy-side memory limits the generality of the proposed metrics and findings, including value smearing.

I also still think the empirical study is somewhat limited in breadth. While the additional Gated DeltaNet result is helpful, my broader concern about the completeness of the evaluation remains, particularly given the absence of Titans and model-based baselines.

That said, after reading the rebuttal and the discussion, I now view these issues primarily as limitations of scope and empirical breadth rather than fatal flaws. The authors clarified that the paper is intentionally focused on model-free RL, and under that framing, I agree that the benchmark and analysis tools provide a useful contribution for studying memory in partially observable model-free RL. For this reason, I am revising my recommendation to weak accept.

**Key Questions For Authors:**

- The paper evaluates several memory architectures, including GRU, transformers, and state-space models. Could the authors provide additional analysis on whether architectural design choices (e.g., attention vs recurrence) systematically influence the degree of value smearing?

**Limitations:**

Please refer to Weaknesses.

**Strengths And Weaknesses:**

Strengths

- Simple testbed for validating memory-based rl algorithms. It also introduces metrics such as Observability Gap and Memory Bias to disentangle memory effects from policy performance.
- The paper is generally well structured and clearly written. Recall density visualizations that explain how agents use memory over trajectories are intuitive.  ￼
- Understanding the role of memory in reinforcement learning under partial observability is an important problem. The proposed tools may provide a useful framework for rigorous evaluation of memory-based RL methods along with the identification of the value smearing pathology and recurrent state contamination effects.

Weaknesses

- POPGym Arcade appears somewhat incremental compared to existing memory-based RL benchmarks such as BSuite [1]. Additionally, POPGym [2] already exists, and extending it with JAX for improved sample efficiency and introducing MDP/POMDP twins seems to be a relatively minor contribution.
- Although the evaluation includes several memory architectures (e.g., transformer, linear transformer, linearized GRU, and GRU), it omits widely studied model-based RL approaches that learn world models capable of compressing historical observations into latent states for policy input. Methods such as Dreamer[3] (recurrent based world model), IRIS[4] (transformer based world model — no memory but should be tested), R2I[5], and mamba world model[6] (SSM based world model), EDELINE[7] (diffusion world model with SSM as memory) should be considered to provide a more comprehensive investigation of memory in reinforcement learning.
- In addition to the evaluated architectures, test-time memory mechanisms should also be examined. Recent approaches such as Gated DeltaNets [8] and Titans [9] represent more advanced linear recurrent architectures and would be valuable to evaluate on the POPGym Arcade.

[1]  Behaviour Suite for Reinforcement Learning. ICLR 2020.

[2] POPGym: Benchmarking Partially Observable Reinforcement Learning. ICLR 2023.

[3] Mastering Diverse Control Tasks through World Models. Nature 2025.

[4] Transformers are Sample-Efficient World Models. ICLR 2023.

[5] Mastering Memory Tasks with World Models. ICLR 2024.

[6] Drama: Mamba-Enabled Model-Based Reinforcement Learning Is Sample and Parameter Efficient. ICLR 2025.

[7] EDELINE: Enhancing Memory in Diffusion-based World Models via Linear-Time Sequence Modeling. NeurIPS 2025.

[8] Gated Delta Networks: Improving Mamba2 with Delta Rule. ICLR 2025.

[9] Titans: Learning to Memorize at Test Time. NeurIPS 2025.

---

> ### Author Rebuttal · Authors · 2026-03-30
>
> We thank the reviewer for recognizing that our proposed tools provide a useful framework for the memory-based RL methods.
>
> > POPGym Arcade appears somewhat incremental compared to existing memory-based RL benchmarks such as ... extending it with JAX for improved sample efficiency and introducing MDP/POMDP twins seems to be a relatively minor contribution.
>
> We created POPGym Arcade because existing benchmarks lack hardware acceleration and unified observation spaces necessary for our memory analysis tools. Computing our metrics required thousands of experiments, making CPU-based environments infeasible; we provide identical MDP and POMDP pixel frames, allowing us to use the exact same network for controlled studies.
>
>
> >  ... it omits widely studied model-based RL approaches...
>
> We agree that investigating memory within model-based architectures, such as world models, is a highly valuable research direction. However, the scope of our current work is focused on diagnosing memory pathologies within model-free reinforcement learning. We will update the title of our paper and the introduction to explicitly specify "Model-Free Reinforcement Learning." Expanding our introspection tools to accommodate the distinct optimization dynamics of world models remains an exciting avenue for future work.
>
> > Recent approaches such as Gated DeltaNets and Titans represent...
>
> We ran experiments on Gated DeltaNets to evaluate test-time memory mechanisms. The results are below:
>
> | Environment | POMDP Return |
> | ----- | ---- |
> | AutoEncodeEasy | 0.26 |
> | BattleShipEasy | 0.74 |
> | BreakoutEasy | 0.47 |
> | CartPoleEasy | 0.99 |
> | CountRecallEasy | 0.35 |
> | MineSweeperEasy | 0.75 |
> | NavigatorEasy | -0.32 |
> | NoisyCartPoleEasy | 0.99 |
> | SkittlesEasy | 0.28 |
> | TetrisEasy | 0.01 |
>
> We will continue to run experiments and update the Memory Bias, Observability Gap, and Recall Density figures to include Gated DeltaNets in our revision. If the paper is accepted, we commit to add Titans architecture experiments in the camera-ready version.
>
> > The paper evaluates several memory architectures, including GRU, transformers, and state-space models. Could the authors provide additional analysis on whether architectural design choices (e.g., attention vs recurrence) systematically influence the degree of value smearing?
>
> We provide a breakdown of the recall density per model and per task in Appendix B. As demonstrated in Figures 12 to 16, value smearing systematically affects all evaluated architectures in both attention-based and recurrent models. We even see this across algorithms (DQN, PPO, Appendices C, D). While the exact shape of the density distributions varies by architecture, we show that regardless of the specific memory model used, prior observations consistently impact future decisions across all MDPs.

---

> > ### Author Rebuttal · Reviewer_9qUy · 2026-04-01
> >
> > While I appreciate the clarification on restricting the scope to model-free RL, my main concern remains unresolved. Memory in reinforcement learning is closely tied to representation learning, and model-based approaches (e.g., world models) are a central paradigm for handling memory through latent state modeling.
> >
> > Focusing only on policy-side memory limits the scope of the study and makes it unclear whether the proposed metrics and findings (e.g., value smearing) generalize beyond this setting. As a result, the contribution appears narrow relative to the broader problem of memory in RL.
> >
> > Additionally, even within this scope, evaluation of recent memory architectures (e.g., Titans) is limited. If the authors could provide experiments with model-based RL on proposed environment and how proposed memory analysis technique can be extended and analyzed compared to model-free RL, I will update the score.

---

### Official Review · Reviewer_H916 · 2026-03-16

**Soundness:** 3
**Presentation:** 3
**Significance:** 2
**Originality:** 3
**Overall Recommendation:** 5
**Confidence:** 4

**Summary:**

This paper introduces a suite of pixel-based environments designed to study memory mechanisms, such as attention-based models, GRUs, and state-space model variants, and their role in learning under both fully observable Markov Decision Processes (MDPs) and partially observable settings (POMDPs). The environments leverage recent advances in JAX to enable efficient parallelization, allowing hundreds of environments to be simulated simultaneously. To evaluate reinforcement learning agents equipped with different memory mechanisms, the authors propose several metrics. These include the observability gap, which measures the performance difference when moving from fully observable MDPs to partially observable POMDPs, and memory bias, which examines how the additional complexity and capacity introduced by memory mechanisms affect performance even in fully observable environments. Their analysis suggests that memory mechanisms can sometimes lead to incorrect value propagation, potentially reducing learning efficiency in POMDP and out-of-distribution settings.

**Compliance With Llm Reviewing Policy:**

Affirmed.

**Final Justification:**

The authors have addressed my concerns satisfactorily.

**Key Questions For Authors:**

1. The authors mentioned that the RL agents struggle to solve POMDP Tetris. Can the authors share some insights on why this is the case?

**Limitations:**

1. The paper includes only environments with discrete action spaces.
2. There are only single agent games. No environments using multi-agents were proposed.

**Strengths And Weaknesses:**

# Strengths:
1. Paper is well written and easy to follow.
2. Figures are clearly presented
3. Equations are clearly defined


# Weakness:
1. Unclear why some games such as POMDP Tetris cannot be solved
2. Conclusion on which memory mechanism is better or more critical for which POMDP environments are not clearly discussed
3. A soft critic: Paper seems a few years late given that most of the RL field have moved on. The authors could provide a short discussion on how the paper aligns with the current state of research in RL.

---

> ### Author Rebuttal · Authors · 2026-03-29
>
> We sincerely thank you for your time and constructive feedback. We will address each point below.
>
> > Insights on why agents struggle with POMDP Tetris.
>
> We discuss this briefly in Section 5 under "Task Diversity and Difficulty" as well as Appendix F. Beyond this:
>
> - State-space MDP tetris is not solved by RL [1,2], our implementation uses pixel observations which are even harder.
> - Even Modern VLMs like GPT-5.2 struggle with **MDP** Tetris: "Top human players still beat certain frontier models consistently." [4]
> - Our POMDP variant is Invisible Tetris, which Tetris Grand Masters also struggle with [3].
> - POMDP Tetris requires perfect memorization [3]. A small error in state tracking quickly compounds and causes failure within couple of timesteps in the future.
>
> > Conclusion of which memory mechanism is better for which POMDP
>
> That's a good point, we updated the Results section to explain this. Under our proposed metrics, the Linear Recurrent Unit (LRU) has the second best Observability Gap the best Memory Bias, implying it will scale best with compute and number of parameters.
>
> Specifically, we find LRU and GRU are both work well in tasks requiring complex memory reasoning over long horizons, such as BattleShip, Breakout, Navigator. The Linear Transformer and Transformer are good at short context tasks like MineSweeper or tasks involving long horizons but simple temporal dependencies like CartPole. We updated Section 6 to explain this.
>
> > Alignment with the current state of research in RL
>
> We do think our conclusions and findings are useful for modern RL. We reveal that out-of-distribution scenarios corrupt agent memory and trigger compounding long-term errors (Fig. 8, 9), explaining one reason why sim-to-real transfer and offline RL remain challenging.
>
> We see similar behavior even with the transformer model. So it is possible these phenomena may also exist to some extent in LLMs doing RLHF. We have added a few sentences of discussion to our manuscript.
>
> > Only discrete action spaces and single agent setting
>
> If accepted, we will add some continuous action tasks. We focused on single-agent POMDPs because PGX [5] and JaxMARL [6] already provide multi-agent games.
>
> We hope we addressed your concerns. Please let us know if any remain.
>
> ### References
> [1] [The Game of Tetris in Machine Learning. Simón Algorta, Özgür Şimşek. Video Games and ML Workshop at ICML.](https://arxiv.org/abs/1905.01652)
>
> [2] [Enhancing Tetris Gameplay with Deep Reinforcement Learning. Dawid Popek. Thesis.](https://www.theseus.fi/handle/10024/863905)
>
> [3] [Invisible Tetris World Championship - Final Match. YouTube.](https://www.youtube.com/watch?v=M6VrvN9zaSE)
>
> [4] [I Built TetrisBench, Where LLMs Compete at Playing Tetris. Here’s What I Found. Yoko Li. Andreessen Horowitz. ](https://a16z.com/i-built-tetrisbench-where-llms-compete-at-playing-tetris-heres-what-i-found/)
>
> [5] [Pgx: Hardware-Accelerated Parallel Game Simulators for Reinforcement Learning. Koyamada et al. NeurIPS.](https://github.com/sotetsuk/pgx)
>
> [6] [JaxMARL: Multi-Agent RL Environments and Algorithms in JAX. Rutherford et al. NeurIPS.](https://github.com/flairox/jaxmarl)

---

> > ### Author Rebuttal · Reviewer_H916 · 2026-04-03
> >
> > I have no other further concerns and have updated my score to 5.

---

### Decision · Program_Chairs · 2026-04-30

**Decision:**

Accept (spotlight)

**Comment:**

This paper introduces an extension of POPGym to Jax and proposes different metrics to characterize the role of memory in an agent's behaviour. Overall, the idea of using PopGym Arcade to study memory mechanisms in model-free RL agents was very well-received. The reviewers were excited about the possibility of mechanistically dissecting RL in POMDPs and how PopGym Arcade could empower the community to gain further insights into the role of memory in POMDPs. Beyond the idea itself and its timeliness, reviewers praised the simplicity of the framework and the paper's presentation. After the rebuttal, the only question left for discussion was the scope of the paper. Eventually, everyone agreed that the paper should be accepted as is, with a focus on model-free RL. All reviewers unanimously agreed that the paper should be accepted.